# Structural and mutational analysis of MazE6-operator DNA complex provide insights into autoregulation of toxin-antitoxin systems

Khushboo Kumari [1] & Siddhartha P. Sarma [1]✉

Of the 10 paralogs of MazEF Toxin-Antitoxin system in *Mycobacterium tuberculosis*, MazEF6 plays an important role in multidrug tolerance, virulence, stress adaptation and Non Replicative Persistant (NRP) state establishment. The solution structures of the DNA binding domain of MazE6 and of its complex with the cognate operator DNA show that transcriptional regulation occurs by binding of MazE6 to an 18 bp operator sequence bearing the TANNNT motif (-10 region). Kinetics and thermodynamics of association, as determined by NMR and ITC, indicate that the nMazE6-DNA complex is of high affinity. Residues in N-terminal region of MazE6 that are key for its homodimerization, DNA binding specificity, and the base pairs in the operator DNA essential for the protein-DNA interaction, have been identified. It provides a basis for design of chemotherapeutic agents that will act via disruption of TA autoregulation, leading to cell death.

[1] Molecular Biophysics Unit, Indian Institute of Science, C. V. Raman Road, Bangalore, Karnataka 560012, India. ✉email: sidd@iisc.ac.in

Toxin-Antitoxin (TA) systems are stress response systems found on plasmids and chromosomes of bacteria and archaea[1–3]. They are involved in important functions such as plasmid maintenance[2,4], programmed cell death[5,6], persister cell formation[7–9], phage abortive infection[10], bacteriostasis and formation of antibiotic-resistant population[11,12]. Based on the mode of action and chemical nature of the antitoxin, TA systems are divided into eight classes, Type I–Type VIII[13]. Among these, Type II system is the most widely represented and also the most extensively characterized[14,15]. According to TADB2.0 database generated in 2017, there are 105 experimentally validated Type II TA pairs[16].

The TA systems in *Mycobacterium tuberculosis (Mtb)* are often attributed to growth, persister cell formation and pathogenicity[14,17]. Under conditions of stress the H37Rv strain of *Mtb* persists in host granulomas in a non-replicating and drug-tolerant state for extended periods of time and causes disease when the stress is relieved. Several free-living bacteria such as *Mtb* and *Nitrosomonas europaea* are known to encode for a large number of TA systems[1]. The H37Rv strain of *Mtb* encodes an exceptionally high number (≈90) of TA systems, while the non-pathogenic mycobacterial species, *M. smegmatis* encodes only 5 TA systems and is relatively fast growing[18,19]. The absence of TA loci in humans and their specific presence in bacteria make them a potential drug target.

An overwhelming majority of the TA systems encoded in the *Mtb* H37Rv genome are of Type II, the prominent members of which are the VapBC and MazEF families that number 50 and 10 each[20]. Generally both the VapBC and MazEF systems are known to inhibit translation by cleaving RNAs. The toxin and antitoxin of the Type II TA systems are polypeptides. Both genes are transcribed under the same promoter. The antitoxin protein forms a complex with toxin, thereby preventing the toxic effects. Under conditions of stress, antitoxins are degraded and toxins are free to attack their cellular targets and help the cell enter a dormant state[13]. Antitoxin also represses the transcription of the operon by binding to the operator region. The toxin acts as a co-repressor at a toxin:antitoxin ratio of 1:1 but at a higher ratio, it acts as a de-repressor (Fig. 1a)[21].

The toxins of the MazEF family in *Mtb, viz.,* MazEF1-MazEF10 are ribonucleases and cleave specific mRNAs and rRNAs. Overexpression of MazF3, MazF6 and MazF9 has been shown to retard cell growth in *M. bovis* BCG and *Mtb* H37Rv and moreover, deletion of these genes make the bacteria more sensitive to antibiotics[22–26].

MazE antitoxins serve to neutralize MazFs and also to repress their operon as binary complexes or ternary TA-operon complexes. Analysis of sequence and available structures of MazE antitoxins have shown them to be composed of two domains: an amino-terminal DNA-binding domain (DBD) and an intrisically disordered carboxy-terminal region involved in toxin binding[27–29].

Structural data are available for MazE4, MazE7 and MazE9 from *Mtb*[30–32], and MazE from E. coli[33,34], *B. subtilis*[35], and *S. aureus*[36].

MazEF6 TA system plays an important role in *Mtb* Non Replicative Persistant (NRP) state establishment. *Mtb* must adapt to nutrient-limiting, hypoxic and other adverse conditions to remain dormant and persist in the host, for which the TA systems are critical. Tiwari et al. have shown that overexpression of MazF3, MazF6 and MazF9 reduced bacterial loads by 8.0-, 22.0- and 18.0-fold, respectively. The overexpression of the remaining MazF homologues, MazF1, MazF2, MazF4, MazF5, MazF7 and MazF8 did not inhibit *M. bovis* BCG growth. Further, the expression level of *mazF6* transcripts have been shown to be even higher than those of *mazF3* and *mazF9* under nitrosative, hypoxic and nutrient stress conditions and upon drug treatment[26]. Thus,

the MazEF6 TA system plays a critical role in the establishment of the NRP state of *Mtb*.

Autoregulation of MazEF6 expression occurs by binding of the MazE6 or MazEF6 complex to its operator DNA 99 bp upstream of its gene[37]. However, the structural, kinetic and thermodynamic factors that govern this regulation are not well characterized for MazEF6 system, or for that matter any of the MazEF systems from *Mtb*. The information on autoregulation of MazEF system comes from homologs of other prokaryotes like *E.coli*[33].

Any meaningful understanding of the TA systems requires extensive knowledge of the relationship between structure and function of this class of proteins. Therefore, it is crucial to probe the interaction of antitoxin with the operator DNA to understand the regulation of the TA system. Furthermore, the possibility of other members of *Mtb* MazEF family to regulate each others function is not well understood.

To this end we have investigated the structure, thermodynamic and kinetic properties of the MazE6 (residue 1–82) antitoxin protein and its interaction with operator DNA. From the solution NMR structural studies of the full length and a C-terminal deletion mutant of MazE6 (nMazE6, residue 1–49), we demonstrate that nMazE6 harbours the Ribbon-Helix-Helix (RHH) DNA binding motif. nMazE6 forms a high affinity complex with an 18 bp sequence bearing the -10 promoter region upstream of the *mazE6* gene. This study provides structural and thermodynamic insights related to the operator binding mode of the protein. An Antitoxin-DNA structure has been modelled using data-driven docking protocols in HADDOCK[38]. In addition, the potential for cross regulation of *mazEF9* operon by MazE6 antitoxin has been explored. The results of these studies are presented below.

## Results

**Solution properties of full length antitoxin MazE6.** MazE6 is highly soluble in aqueous buffers and refolds easily upon chemical denaturation. CD spectrum contains signature negative bands of both alpha helix (208 and 222 nm) and beta strand (218 nm) (Supplementary Fig. S1a). Examination of analytical size exclusion chromatogram showed that the protein has an elution volume similar to that of a 27.4 kDa globular protein (Supplementary Fig. S1c). Size Exclusion Chromatography-Multi Angle Light Scattering (SEC-MALS) confirmed that the protein is a dimer (Observed mol. wt. = 17 kDa) in solution (Supplementary Fig. S2). The one-dimensional proton NMR spectrum is resolved and has upfield shifted lines at −0.15 and −0.18 ppm suggesting that the protein is folded (Supplementary Fig. S1b). The two-dimensional $^1$H - $^{15}$N HSQC NMR spectrum of free MazE6 shows several well resolved resonances with narrow line widths while the center of the spectrum (7.8–8.6 ppm) shows resonance lines of high intensity that are significantly overlapped (Fig. 1b). This suggests that the protein has a well-folded region as well as an intrinsically disordered region.

Attempts to obtain sequence specific resonance assignments on uniformly $^{13}$C,$^{15}$N labeled proteins were unsuccessful. Hence backbone assignments were obtained from a suite of triple resonance experiments on perdeuterated protein samples that were $^{13}$C, $^{15}$N labeled with and without ILV-methyl protonation[39]. The sequence specific assignments for select residues obtained from an analysis of HNCACB and HN(CO)CACB data sets are shown in Supplementary Fig. S3a. Figure 1b shows the sequence-specifically assigned $^1$H-$^{15}$N HSQC spectrum of MazE6. Seventy-five of the eighty-one NH resonances (82-1 proline) were assigned unambiguously in the 2D $^1$H-$^{15}$N HSQC spectrum of MazE6. Residues for which the assignment is missing are: M1, S27, K32, Y37, E40 and N71 . Chemical shift assignments

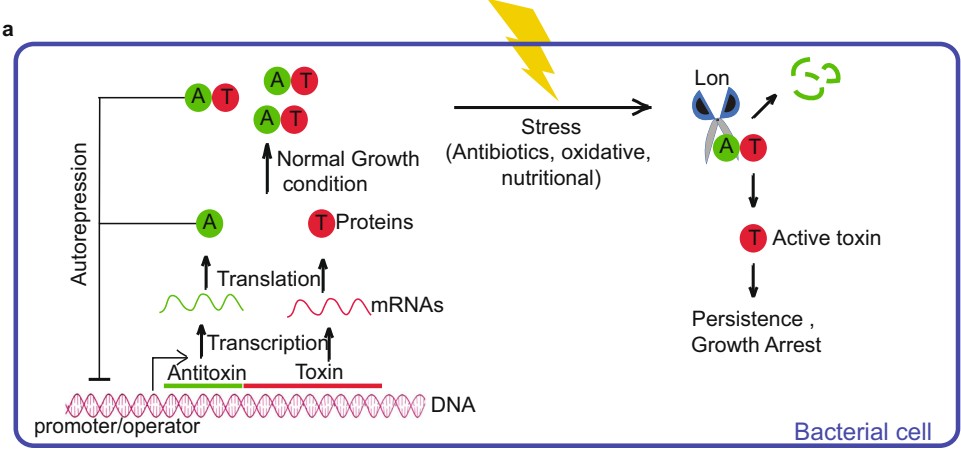

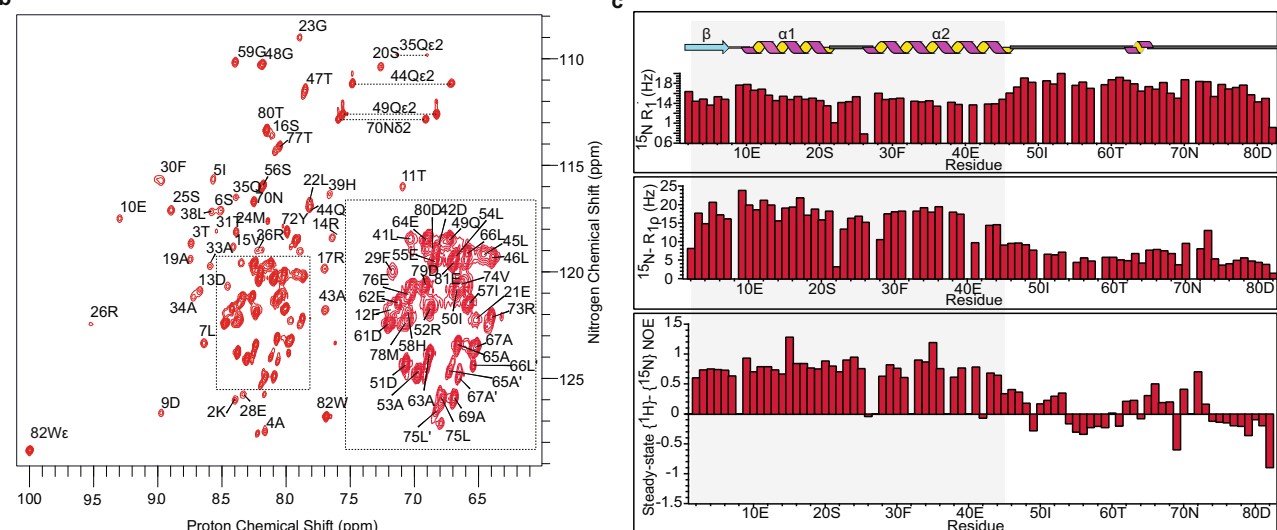

**Fig. 1 NMR characterization of MazE6. a** Schematic representation of working mechanism of Type II TA system. **b** Sequence specific assignments are indicated on the 2D $^1H$-$^{15}N$ HSQC spectrum of MazE6. Inset shows the assignment for the crowded region of the spectrum within the box. Assignments with a prime superscript (') represents doubling of peaks due to conformational heterogenity. Horizontal dotted lines connect resonances of side chain amide protons. See text for details. **c** $R_1$, $R_{1\rho}$ relaxation rates and steady-state heteronuclear {$^1H$} - $^{15}N$ NOEs for $^{15}N$ backbone nuclei of MazE6, plotted against residue number. Low $R_{1\rho}$ relaxation rates and steady-state {$^1H$} - $^{15}N$ heteronuclear NOE values for the C-terminal residues further confirm its disordered (dynamic) nature in solution. Blank spaces correspond to residues for which data is missing. Source data are provided in the Supplementary Data 1.

were possible for 93.75% $H^N$, 70.17% $^{15}N$, 100% $^{13}C^\alpha$, 75.6% CO and 98.7% $^{13}C^\beta$ of the backbone and side-chain nuclei respectively. The sequence information of the protein is provided in Supplementary Fig. S1d.

The sequence specific assignments obtained from triple resonance NMR experiments were corroborated by assignment of sequential, short and medium-range NOEs observed in 3D $^{15}N$-edited NOESY-HSQC spectrum. Secondary structure of MazE6 was calculated from chemical shifts (CSI) of $^1H^\alpha$, $^{13}C^\alpha$, $^{13}C^\beta$ and $^{13}C'$, sequential and short-range NOEs, and backbone $^3J_{H_N H_\alpha}$ coupling constants (Supplementary Fig. S3b). The secondary structure showed that the protein is divided into two domains: a structured N-terminal domain (M1-L46; the putative DNA binding domain) and an intrinsically disordered C-terminal domain (T47-W82; putative toxin binding domain). The extended disorder in the C-terminal region lengthens the hydrodynamic radius, causing the protein to behave like a 27.4 kDa protein in solution. The N-terminal domain is composed of one beta strand (M1-S6) and two helices (D9-E21, R26-L46) (RHH motif) that are connected by loops L1 (L7-P8)

and L2 (L22-S25). The ordered and disordered nature of the N- and C-terminal domains of MazE6 also manifest in the relaxation properties of backbone $^{15}N$ nuclei in the two regions. The average $R_{1\rho}$ for N-terminal region is high (16.41 Hz) as compared to C-terminal region (6.13 Hz). A stark difference in the steady-state heteronuclear {$^1H$} - $^{15}N$ NOEs for the N-terminal and C-terminal residues also shows the structured and disordered nature (Fig. 1c). The HSQC spectrum of MazE6c (G48-W82) exhibits a narrow chemical shift dispersion (Supplementary Fig. S4a) and overlays with the disorderd region of MazE6 (Supplementary Fig. S4b) providing further confirmation that the C-terminal region of MazE6 is intrinsically disordered.

Since the DNA binding region of the MazE6 is a well-folded domain, the structural studies of the interaction of MazE6 with the operator DNA were carried out using a c-terminal deletion mutant of MazE6, henceforth referred to as nMazE6 (Supplementary Fig. S1d).

**Solution properties of nMazE6.** Like MazE6, nMazE6 exists as a stable dimer (Supplementary Fig. S5a) suggesting that the

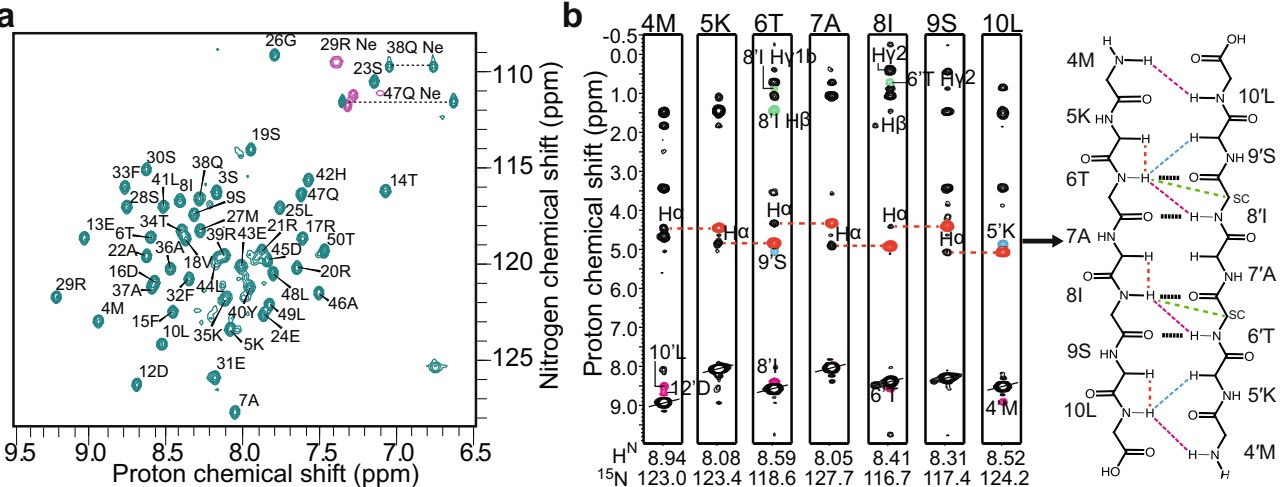

**Fig. 2 Solution properties of nMazE6. a** Assigned 2D $^1$H-$^{15}$N HSQC spectrum of nMazE6. Sequence specific assignments are indicated on the spectrum. Horizontal dotted lines connect resonances of side chain amide protons. **b** Strip-plots from $^{15}$N-edited NOESY data showing sequential intra-subunit and cross-strand long-range NOEs observed across the dimer interface. NOEs are colored black (intra-residue), red (intra-subunit $_{i-1}$H$^α$- $_i$H$^N$), cyan (inter-subunit H$^α$-H$^N$), pink (inter-subunit H$^N$-H$^N$) and green (inter-subunit side chain-H$^N$). β-sheet topology constructed from the pattern of assigned NOEs is shown on the side. NOEs are indicated by colored broken lines. Proposed hydrogen bonding between donor-acceptor pairs in nMazE6 are indicated by black dotted lines.

interface for homodimerization of the protein is present in this domain. The upfield shifted resonances at −0.15 ppm and −0.18 ppm are also present (Supplementary Fig. S5b) indicating that the overall fold of nMazE6 is identical to that of the same domain in MazE6. Deletion of the intrinsically disordered C-terminal domain leads to a well resolved 2D correlation spectrum (Fig. 2b) when compared to the full length protein (Supplementary Fig. S6).

**Solution structure and dynamics of nMazE6.** Sequence specific resonance assignments of backbone and side chain $^1$H, $^{13}$C and $^{15}$N nuclei were accomplished from analysis of NMR data acquired on uniformly $^{13}$C, $^{15}$N labeled samples of nMazE6. Chemical shift assignments were possible for 96% H$^N$, 92.3% $^{15}$N, 96.15% $^{13}$C$^α$, 90.38% CO and 95.9% $^{13}$C$^β$ of the backbone and side-chain nuclei respectively. Also assigned were the residues that were missing in the data for the full-length protein. The $^1$H, $^{15}$N and $^{13}$C assigned resonances of MazE6 and nMazE6 have been deposited in the BioMagResBank (http://www.bmrb.wisc.edu/) under accession number 51226 and 51227 respectively. The secondary structure computed for nMazE6 from $^{13}$C CSI and from sequential-, short- and medium-range NOEs assigned in homonuclear NOESY and isotope-edited NOESY-HSQC spectra, was found to be the same as for the N-terminal region of MazE6 i.e, a RHH (Supplementary Figs. S3b and S5c).

NOE correlations[40] were assigned in a 3D-$^{13}$C and $^{15}$N "mother" NOESY-HSQC spectrum and two-dimensional $^1$H-$^1$H NOESY spectra acquired at 600 MHz ($τ_m = 125$ ms). Structures were calculated with the understanding that nMazE6 exists in solution as a dimer. The intra-subunit restraints were duplicated and renumbered to generate restraints for both subunits. A total of 1190 unambiguous NOEs were assigned for nMazE6. Of these 1056 NOEs and 134 NOEs were assigned as intra-subunit and inter-subunit respectively. Backbone dihedral angles were assigned from a combined analysis of $^{13}$C secondary CSI and sequential-, short- and medium-range NOEs assigned in NOESY spectra. A total of 95 $φ$ and 94 $ψ$ dihedral angles were assigned and incorporated in the structure calculation. Knowledge of secondary structure[40] was used to generate the hydrogen bond

restraints. The details of number and type of restraints used for structure calculation and the structural statistics derived for the ensemble of twenty low energy structures are given in Table 1.

Each subunit of the dimeric molecule in nMazE6 structure possesses the RHH motif (β1, residues M4-L10; α1, residues D12-E24; and α2, residues R29-L49). A superposition of 20 lowest-energy conformers is shown in Fig. 3a. For the residues 5–50 and 5′–50′, average backbone and average heavy atom RMSD to mean is 0.44 and 1.0 respectively. Good stereochemical quality of the structures is reflected from the high percentage of backbone dihedral angles lying in the most favored (95.4%) and additionally allowed regions (4.6%) of the Ramachandran map. Structural coordinates for nMazE6 have been submitted in the PDB with the accession number 7WJ0.

The quaternary structure of nMazE6 (Fig. 3b) shows that dimerization occurs through the N-terminal residues of each subunit via formation of a two-stranded anti-parallel β-sheet. Inter-strand NOEs were crucial restraints for establishing the strand registry (Fig. 2b). Two RHH motifs, one from each monomer intertwine together to form a stable domain having a two-fold symmetry. The β-sheet contains alternating hydrophilic and hydrophobic residues. Hydrophobic residues point inwards and interact with aromatic residues to form the hydrophobic core. 8I and 10L from β1, 15F, 18V and 22A from α1 and 32F, 33F, 36A and 40Y from α2 form the hydrophobic core. CH $\bullet\bullet\bullet$ π and π $\bullet\bullet\bullet$ π interactions are critical to establish this hydrophobic core. CH $\bullet\bullet\bullet$ π interactions between 18V$γ_2^{CH_3}$ and 32F and 8Iδ$_1^{CH_3}$ and 33′F results in upfield shifted lines at −0.15 and −0.18 ppm respectively (Fig. 3d). An aromatic cluster formed by residues across the interface contribute significantly to dimer formation. It has a series of edge-to-face π $\bullet\bullet\bullet$ π stacking involving residues alternating from each subunit forming a donut like aromatic cluster (Fig. 3e). The residues involved are 40Y-32′F-33F-33′F-32F-40′Y, with 33F-33′F having a face to face stacking interaction. The buried surface area and the interface area formed upon dimerization are 3431.7 and 1715.85 Å$^2$ respectively.

Figure 3g shows the measured $^{15}$N $R_1$ and $R_{1ρ}$ relaxation rates and Steady-state Heteronuclear NOE values for nMazE6 as a function of residue number. The positive Steady-state Heteronuclear NOE values suggest that nMazE6 is well ordered in

**Table 1 NMR restraints and structural statistics for nMazE6.**

**NMR distance and dihedral constraints**

Distance constraints

| | |
|---|---|
| Total[a] | 1190 |
| Intra - residue ($\lvert i - j \rvert = 0$) | 376 |
| Inter - residue | |
| Sequential ($\lvert i - j \rvert$ 1) | 322 |
| Medium-range ($\lvert i - j \rvert \leq 4$) | 194 |
| Long-range ($\lvert i - j \rvert \geq 5$) | 50 |
| Inter-subunit | 134 |
| Hydrogen bond restraints | 114 |
| Total dihedral angle restraints | 189 |
| $\phi$ | 95 |
| $\psi$ | 94 |

**Structure statistics**

Violations (mean and s.d)

| | |
|---|---|
| Distance constraints > 0.1Å | 0 |
| Dihedral angles > 5° | 0 |
| Max. Dihedral angle violation (°) | 0 |
| Max. Distance constraints violation (Å) | 0 |
| van der Waals violations | 5 |
| Deviations from idealized geometry | |
| Bond lengths (Å) | 1 |
| Bond angles (°) | 37 |
| Impropers (°) | 35 |

**Ramachandran map statistics**

*PROCHECK*

| | |
|---|---|
| Most favored regions (%) | 95.4 |
| Additionally allowed regions (%) | 4.6 |
| Generously allowed regions (%) | 0 |
| Total allowed regions (%) | 100 |

*Richardson MolProbity*

| | |
|---|---|
| Favored regions (%) | 98.2 |
| Allowed regions (%) | 1.1 |
| Total allowed regions (%) | 100 |
| Disallowed regions (%) | 0.7 |
| Average pairwise r.m.s deviation Å)[b] | |
| Heavy | 1.0 ± 0.2 |
| Backbone | 0.44 ± 0.09 |

[a]Total number of restraints used for the structure calculation of the dimer.
[b]Pairwise r.m.s. deviation was calculated among 20 refined structures.

solution. The relaxation rates of residues in nMazE6 are similar to those found for residues in N-terminal region of MazE6. The similarity in the relaxation parameters indicate that the nMazE6 have the same structural and dynamic properties as that of N-terminal region of MazE6 and that nMazE6 is an independently folded "domain". Analysis of the electrostatic surface potential (Fig. 4a) of the nMazE6 structure reveals that the solvent exposed face of the $\beta$-sheet is dominated by positive potential while the distal C-terminal tail of nMazE6 has predominantly negative potential.

**MazE6-operator binding study.** The operator sites of TA systems in mycobacteria are usually centered at −10 promoter region and often overlap with the regulatory elements recognized by RNA polymerase[41,42]. Therefore, a stretch of 18 bp upstream of *mazEF6* Transcription Start Site (TSS)[43] which includes the −10 promoter region was selected to probe the protein-DNA interaction using biophysical methods (Gel-filtration, Isothermal calorimetry and NMR titration). The sequence of the sense and antisense strands are 5′ -CCG GTT ATA CTA TCT GTA-3′ and 5′ -TAC AGA TAG TAT AAC CGG-3′ respectively.

An overlay of gel filtration chromatograms of MazE6, operator DNA, and MazE6+DNA (in the ratio 1:2) are shown in Fig. 4b. Comparision of the chromatograms indicate that a complex

(Peak 2) is formed which elutes at a retention volume lower than that of free DNA and free protein. NMR spectroscopy of Peak 2 demonstrates the presence of both MazE6 (presence of indole protons at 10 ppm) and DNA (presence of characteristic imino protons between 12 and 14 ppm) proving the formation of complex (Fig. 4c). The experiment was repeated with nMazE6 and a clear formation of complex is seen (Supplementary Fig. S7a) in this case too. This proves that the interaction is mediated through the N-terminal domain. Also, the electrostatic potential of the N-terminal region is positive (Fig. 4a) suggesting that the DNA binding motif lies in the N-terminal region. To show that the interaction is sequence dependent and specific to its operator sequence, the operator DNA sequence was scrambled and then used to study complex formation. The elution volumes of individual components did not change showing that complex had not formed (Supplementary Fig. S7b).

Details of antitoxin and operator DNA interaction at atomic resolution were ascertained from NMR chemical shift perturbation studies. Chemical shift mapping studies were carried out for both MazE6 (Supplementary Fig. S8a) and nMazE6 (Fig. 5a and Supplementary Fig. S8b). Changes in chemical shift were observed for several residues located in the secondary structural elements in each subunit over the range of molar ratios used for the study. With increasing concentration of DNA, residues in $\beta1$, $\beta1'$, $\alpha2$ and $\alpha2'$ exhibited severe line broadening, to the point of disappearance at a molar ratio of 0.2. The resonance lines of these residues reappear at a higher molar ratio. This is indicative of an intermediate to a slow intermediate exchange regime. Figure 5c and d show a plot of the magnitude change in chemical shift ($\Delta\delta$ ppm) as a function of primary structure for nMazE6 and MazE6, respectively. Residues in the C-terminal region of MazE6 did not show any significant chemical shift perturbation indicating that they are remote from the binding site. The specific residues involved in the interaction are mapped onto the structure (Fig. 5e).

The G-X-S/T/N motif present in the loop L2 is conserved in MazE6 as well (Fig. 3f) in the form of G-M-S. It allows the correct positioning of the N-terminus of the $\alpha2$ helix[44] and helps in interaction with the DNA. To test for specificity of interaction, CSP studies were carried out using operator DNA of the *mazEF9* operon. In this case, a continuous shift of resonance lines without any significant line broadening was observed, indicative of fast exhange and weak binding (Supplementary Fig. S9a). The dissociation constants were calculated from the titration using TITAN software[45]. TITAN quantitatively models the titration data by numerically integrating the Bloch-McConnell equations at each titration points and directly fits the peak lineshape observed in the corresponding HSQC spectrum. Of the residues which showed chemical shift perturbation >0.05 ppm ($\Delta\delta$), the shifts of 24 residues were unambiguously tracked during the course of titration and were used to calculate the kinetic and thermodynamic properties of the intermolecular interaction. $K_d$ values thus obtained were 0.09 $\mu$M for cognate *mazEF6* operator DNA and 20.6 $\mu$M for non-cognate *mazEF9* operator DNA and the corresponding $k_{off}$ rates being 125 and 1696 per second respectively. The calculated $k_{on}$ rates are $147 \times 10^7$ and $8.2 \times 10^7$ M$^{-1}$ s$^{-1}$ respectively. The values of these rate constants for cognate and non-cognate interactions are tabulated in Table 2. There is excellent agreement between experimentally observed lineshapes and simulated lineshapes, confirming the reliability of the fit and the parameters derived from it (Supplementary Fig. S10).

Thermodynamic parameters of the antitoxin and operator DNA interaction which includes equilibrium dissociation constant ($K_d$), enthalpy change ($\Delta H$), entropy change ($\Delta S$), and stoichiometry ($n$) were extracted from Isothermal Titration

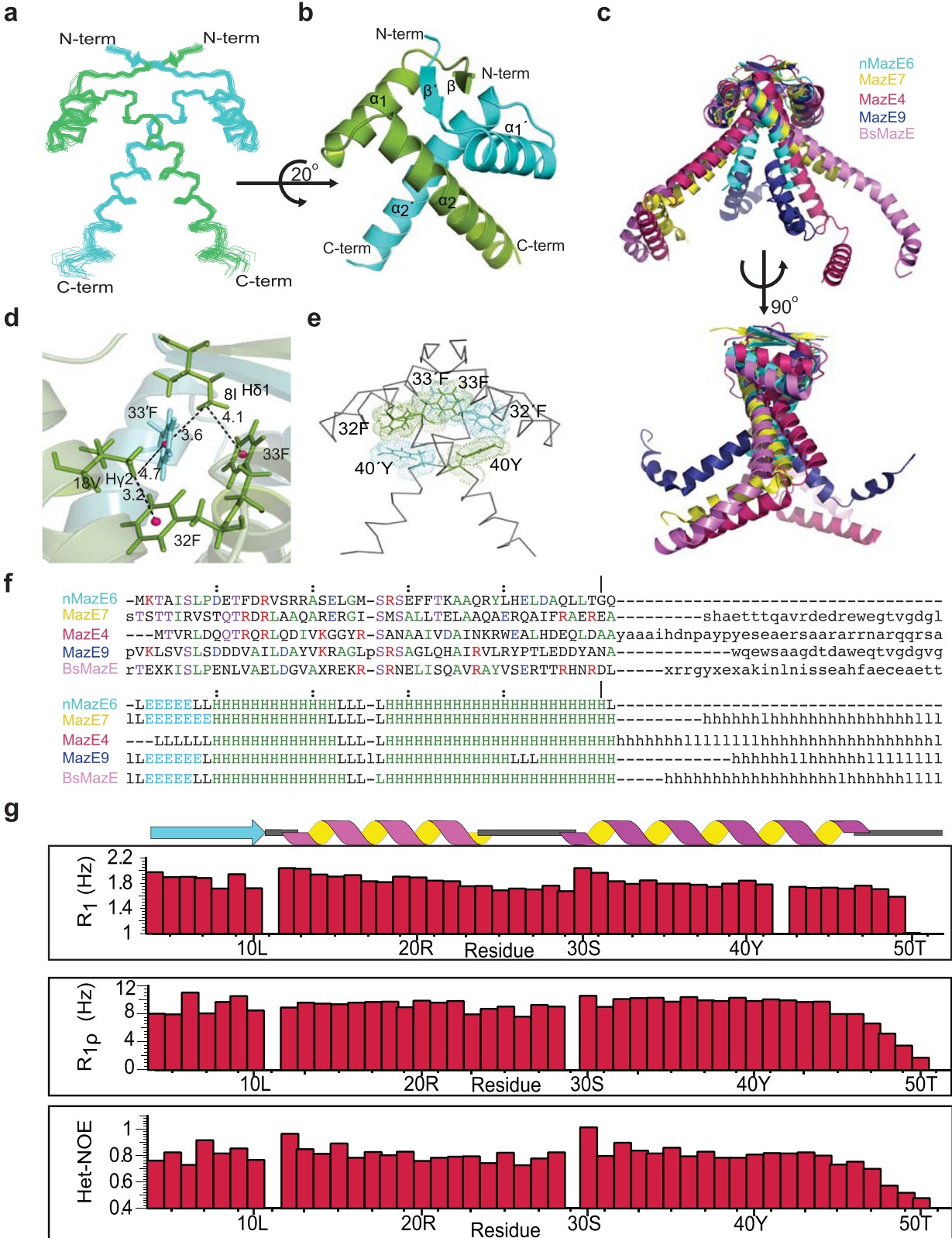

Calorimetry (ITC) data. The interaction of nMazE6 with its cognate DNA is exothermic in nature with a dissociation constant of 0.22 $\mu$M and stoichiometric value(n) of 0.93 indicating an 1:1 interaction (Fig. 6a). Table 2 shows the results obtained from ITC experiments. No heat change was observed for the interaction of nMazE6 with scrambled DNA and a very small heat change

accompanied titration with non-cognate *mazEF9* operator DNA for which $K_d$ could not be determined (Fig. 6c and d), indicating that the antitoxin binding is specific to its cognate operator DNA. The results of the NMR and ITC titration studies unequivocally establish that MazE6 regulate its cognate operon via a highly specific and strong interaction.

**Fig. 3 Quaternary structure and dynamics of nMazE6. a** Ensemble of twenty lowest energy structures calculated for nMazE6 emphasizing the structural symmetry of the dimer. The structures have been superposed on backbone N, $C^\alpha$ and $C'$ atoms for residues 4–52 and 4'–52'. **b** Cartoon representation of the lowest energy structure. The secondary structural elements are labeled in order of sequence in each subunit. The figure was generated in PyMol using the structural coordinates of nMazE6. **c** Alignment of MazE homologs: nMazE6 (cyan), MazE7 (yellow), MazE4 (magenta), MazE9 (navy) and BsMazE (mauve), showing the difference between them. **d** CH •••  $\pi$ interactions between $18V\gamma_2^{CH_3}$ and 32F and $8l\delta_1^{CH_3}$ and 33' F resulting in upfield shifted lines at −0.15 and −0.18 ppm (Supplementary Fig. S5b) respectively. The distance between the methyl groups and the aromatic ring centroid (pink) is indicated by dashed line. **e** Aromatic cluster formed by 40Y-32' F-33F-33' F-32F-40' Y. **f** Structure based sequence alignment of nMazE6 with other MazE from *Mtb* and *Bacillus subtilis* MazE. PDB ID: nMazE6 (7WJ0), MazE7(6A6X), MazE4 (5XE3), MazE9 (6KYT) and BsMazE (4ME7) respectively. Structural alignment was done using the DALI server[68](**g**) $R_1$, $R_{1\rho}$ relaxation rates and steady-state heteronuclear {$^1$H} - $^{15}$N NOEs for $^{15}$N backbone nuclei of nMazE6, plotted against residue number. Source data are provided in the Supplementary Data 1.

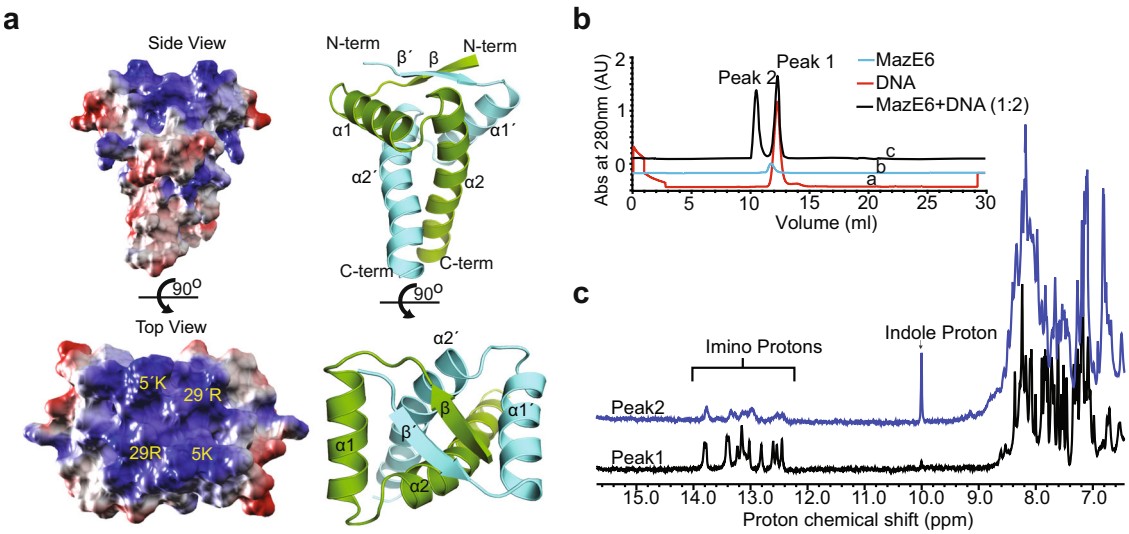

**Fig. 4 Interaction of nMazE6 with operator DNA. a** Side view and top view of surface charge potential of nMazE6 . The molecular surface having positive potential is shown in blue and negative potential is shown in red. Electrostatic surface was calculated and rendered using the Adaptive Poisson-Boltzmann Solver package within PyMOL. **b** Stacked plot of gel-filtration chromatograms of operator DNA (red), MazE6 (blue) and MazE6+DNA (1:2) (black) on Superdex-S75 analytical column (24 mL). Comparison of the chromatograms shows that Peak1 and Peak2 in chromatogram (**c**) corresponds to DNA and MazE6-DNA complex respectively. Absorbance axis has been offset for clarity. **c** 1D Proton NMR spectra of analytes in Peak 1 and Peak 2. Molecular species present in Peak 1 corresponds to DNA (presence of characteristic imino protons between 12 and 14 ppm) and in Peak 2 corresponds to DNA + protein (presence of imino between 12 and 14 ppm and indole protons at 10 ppm, respectively). Line broadening of resonance line of imino protons in Peak 2 are indicative of complex formation. Similar profiles are observed for interaction between nMazE6 and DNA (Supplementary Fig. S7a). Source data are provided in the Supplementary Data 1.

**HADDOCK model of MazE6-operator complex**. The quaternery structure of nMazE6 shows clear separation of electrostatic potential where the positive potential is concentrated at the N-terminus and a negative potential at the C-terminus of the protein (Fig. 4a). NMR chemical shift perturbation data clearly shows that the residues at the N-terminus are responsible for DNA binding. To gain insight into the structural mechanism by which nMazE6 binds to the operator DNA, data driven docking protocols in HADDOCK were applied[38]. Structural restraints obtained from CSP data were input as Ambiguous Interaction Restraints. Residues of nMazE6 that showed chemical shift perturbation more than average plus one standard deviation were found to be present in the positively charged surface and were considered as active residues (Refer Materials and methods).

HADDOCK clustered 181 structures (total of 200) into 7 clusters which represents 90.0% of the water-refined models HADDOCK generated. Of the 7 clusters, Cluster 1 represented the best structures with lowest Z-score and HADDOCK score of −1.2 and −106.9 ± 1.6 a.u. respectively and a cluster size of 68. The high quality of the docked structures of the complex are reflected in the low RMSD value, large buried surface area and favorable energetics of interaction. Structural statistics for the

models in Cluster 1 are given in Table 3. Structural coordinates for nMazE6-operator DNA complex have been submitted in the PDB with the accession number 7WNR.

From the structures of the complex (Fig. 7a), it is clear that the $\beta$-sheet of nMazE6 is anchored in the major groove of the DNA. Residues 5K, 5' K, 7A and 7' A make specific base contacts in the major groove. In addition, important electrostatic interactions ($E_{Elec}$ = −240.8 kcal/mol) between the sugar-phsphate backbone and the side chains of 4M, 5K, 15F, 28S, 29R and 30S from each subunit contribute to stabilization of the complex (Fig. 7b and c). It is evident from the complex structure that NNN position in the TANNNT (TATACT) nucleotide sequence is responsible for specific base contacts with the protein. To confirm the role of NNN nucleotides, the operator DNA was mutated at these positions from 5' -CCGGTTATACTATCTGTA-3' to 5' -CCGGTTAGCAT ATCTGTA-3'. Chemical shift perturbation study showed a shift of interaction regime from slow exchange to fast exchange (Fig. 5a, b and Supplementary Fig. S9b) and that the interaction had weakened 100 fold for the mutated operator DNA ($K_d$ = 8.5 $\mu$M) as compared to the wild type ($K_d$ = 0.09 $\mu$M). ITC titration study indicated a very small heat change and a $K_d$ could not be determined (Fig. 6b) for this interaction, thereby corroborating the NMR observations.

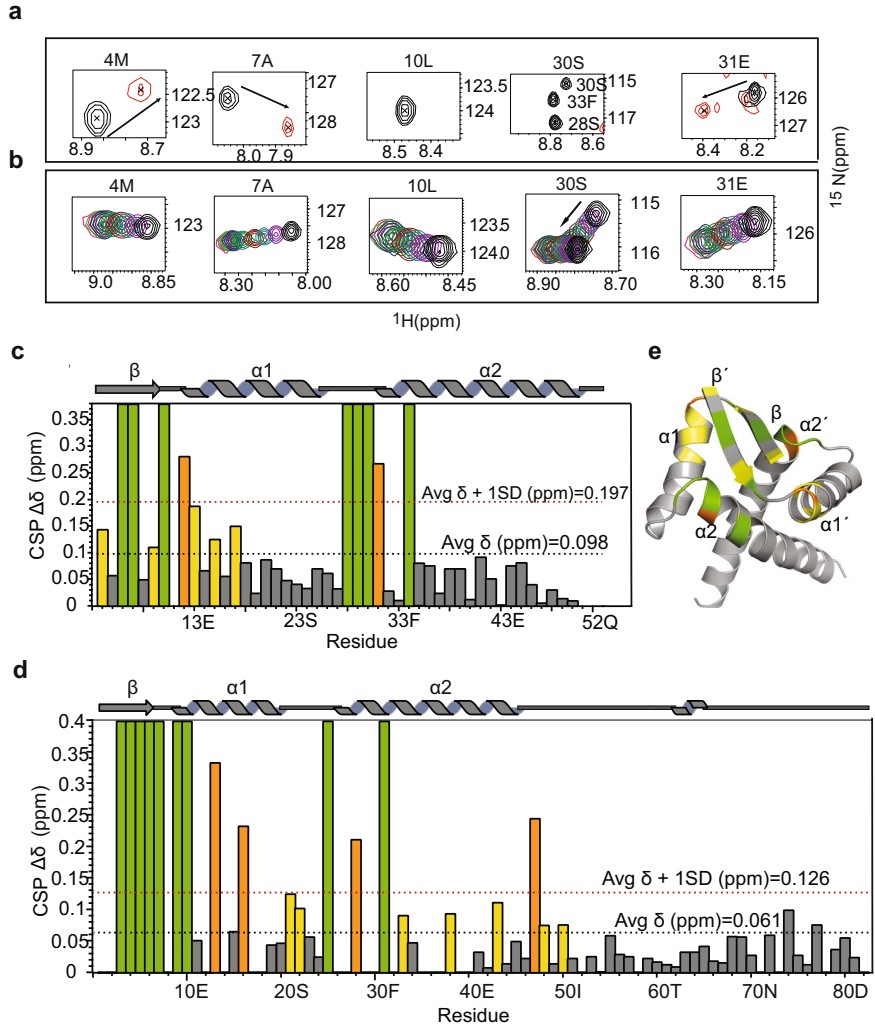

**Fig. 5 Chemical Shift Perturbation (CSP) upon operator DNA binding. a** Chemical shift perturbations observed for few residues in the $^1$H-$^{15}$N HSQC (Supplementary Fig. S8b) of nMazE6 upon titration with cognate operator DNA at a 1:2 (nMazE6:DNA) molar ratio. Spectra of free nMazE6 (Black) and nMazE6 in complex with operator DNA at a protein:DNA ratio of 1:2 (Red) are shown. Changes in chemical shifts of backbone amide resonance position between the first and last titration points are indicated by black arrows. For CSP of MazE6, refer Fig. S8A. **b** Chemical shift perturbations of key residues observed in the $^1$H-$^{15}$N HSQC of nMazE6 upon titration with *mazEF6* mutated operator DNA. For full spectrum, refer Supplementary Fig. S9b. **c** Plot of cumulative CSP as a function of the primary structure of nMazE6. **d** Plot of cumulative CSP as a function of the primary structure of MazE6. Average CSP and average CSP plus one standard deviation are indicated by dotted lines (Black) and (Red), respectively. **e** Residues that showed above-average CSP (yellow), above-average CSP + 1SD (orange), and residues that completely broadened and hence disappeared upon DNA binding (green) are mapped on the structure of nMazE6. Source data are provided in the Supplementary Data 1.

**Table 2 Thermodynamic and kinetic parameters derived for interaction of nMazE6 with operator DNA constructs.**

**Thermodynamic parameters derived from ITC experiments for interaction of nMazE6 with different DNA**

| S. No. | Experiment | $K_d$ ($\mu$M) | $\Delta$G (kcal/mol) | $\Delta$H (kcal/mol) | T$\Delta$S (kcal/mol) | $n$ |
|---|---|---|---|---|---|---|
| 1. | nMazE6-*mazEF6* operator DNA | 0.29 ± 0.04 | −9.08 ± 0.22 | −6.13 ± 0.15 | 2.95 | 0.93 ± 0.01 |
| 2. | nMazE6-*mazEF6* operator DNA | 0.22 ± 0.02 | −9.24 ± 0.11 | −6.49 ± 0.08 | 2.76 | 0.94 ± 0.01 |
| 3. | nMazE6- *mazEF6* mutated DNA | – | – | – | – | – |
| 4. | nMazE6-scramble DNA | – | – | – | – | – |
| 5. | nMazE6-*mazEF9* operator DNA | – | – | – | – | – |

**Thermodynamic and kinetic parameters derived from NMR titration experiments**

| | Experiment | $K_d$ ($\mu$M) | $k_{on}$ (M$^{-1}$s$^{-1}$) | $k_{off}$ (sec$^{-1}$) |
|---|---|---|---|---|
| 1. | nMazE6-*mazEF6* operator DNA | 0.09 ± 0.01 | (147 ± 22) × 10$^7$ | 125 ± 4.04 |
| 2. | nMazE6-*mazEF6* mutated DNA | 8.59 ± 0.31 | (17.30 ± 0.93) × 10$^7$ | 1486.44 ± 27.70 |
| 3. | nMazE6-*mazEF9* operator DNA | 20.60 ± 1.81 | (8.2 ± 0.94) × 10$^7$ | 1696 ± 47.14 |

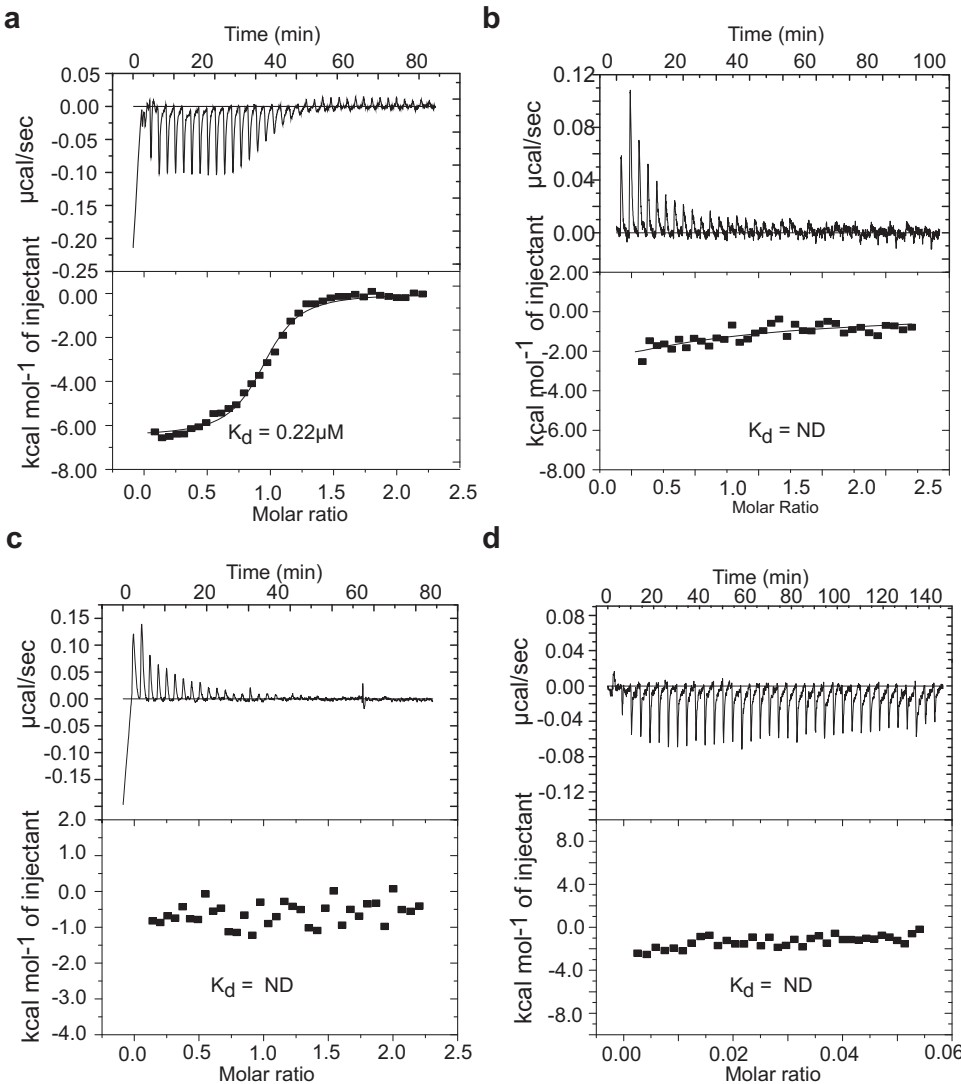

**Fig. 6 Isothermal titration calorimetry of nMazE6-operator DNA binding. a** Isothermogram of the titration of operator DNA with nMazE6 at 30 °C. An exothermic interaction was observed with a $K_d$ of 0.22 $\mu$M. **b** Isothermogram of mutated operator DNA at 30 °C. **c**, **d** Isothermograms of the titration of scrambled DNA and non-cognate (*mazEF9*) operator DNA with nMazE6 at 30 °C respectively. The top portion of each panel shows the raw titration data, and the bottom portion of each panel shows the integrated heats fit to single-binding site isotherms after subtracting the heats of dilution (endothermic in this case) (Refer Materials and method for details). The calculated ($K_d$) values are shown in each panel. The concentration of the titrate (DNA) was 10 $\mu$M, while that of the titrant (nMazE6) was 100 $\mu$M. (ND- Not determined). Source data are provided in the Supplementary Data 1.

## Discussion

The functional studies of mycobacterial MazEF systems suggest that MazEF6 along with other TA systems play an important role in *Mtb* persister cell formation[26]. The structures of the MazEF4, MazEF7 and MazEF9[30–32] provide detailed information on the structural mechanism by which the antitoxins neutralize their cognate/non-cognate toxins. Structural data for the Toxin-substrate complex is scarce. To date, data are available only for the Toxin-substrate RNA complexes of MazF6 (PDB ID 5HK0, 5HKC, 5HK3) and MazF9 (5HJZ). Importantly, the details of the structural basis of autoregulation of *Mtb* MazEF has not been reported.

The present study was carried out to structurally characterize the MazE6 antitoxin polypeptide and its function as a transcriptional regulator. MazE6 is a dimer in solution and harbors an RHH domain at the N-terminus similar to those observed for the homologous antitoxins, *viz.*, MazE4, MazE7 and MazE9 despite a low sequence identity (19%, 17% and 22% respectively) with the latter. The RHH motif was first observed in the Arc-repressor

protein[46] thus establishing a unique transcription factor superfamily. Other Type II systems having the same domain are *Bacillus subtilis* MazE, and E. coli Relb, ParD and CcdA[47–49]. Structural alignment of nMazE6 with the above mentioned antitoxins shows that L10 in the loop L1, G26-M27-S28 in the loop L2 and R29 in the helix $\alpha$2 are conserved. These residues help in the optimal positioning of the helices for interaction with the DNA. The extent of stabilization that is provided by the cluster of CH $\bullet\bullet\bullet$ $\pi$ and $\pi$ $\bullet\bullet\bullet$ $\pi$ interactions to the dimerization interface of MazE6 is not seen in any of the structures of the homologs (MazE4, MazE7 and MazE9). While, MazE4, MazE7 and MazE9 have two, one and two aromatic residues in their N-terminal domain respectively, MazE6 has four of them (Fig. 3c and f). Four aromatic residues from each monomer make a donut like aromatic cluster which stabilizes the protein and makes a sort of hydrophobic core of the protein. (Fig. 3e). Thus, the N-terminal region is structurally rigid.

On the other hand, the C-terminal region of MazE6 is intrinsically disordered in the free form, except for a small helix

**Table 3 Structural statistics for data-driven docking of nMazE6 and operator DNA.**

| | |
|---|---|
| **Ambiguous interaction restraints** | |
| nMazE6 | 5,7,9,15,29,30, 5′,7′,9′,15′,29′,30′ |
| operator DNA | 5,6,7,8,9,10,11,12,25,26,27,28,29,30,31,32 |
| **Restraints for nMazE6** | |
| NOE | 1190 |
| H-bond | 114 |
| Dihedral angle | 189 |
| **Structural statistics for haddock-derived structures of nMazE6-DNA complex** | |
| HADDOCK score | −106.9 ± 1.6 a.u. |
| Cluster Size | 68 |
| RMSD | 1.8 ± 1.8 Å |
| $E_{vdw}$ | −61.4 ± 1.4 kcal/mol |
| $E_{EE}$ | −240.8 ± 11.0 kcal/mol |
| $E_{DE}$ | 20.7 ± 1.1 kcal/mol |
| $E_{RVE}$ | 26.6 ± 14.1 kcal/mol |
| BSA | 1383.6 ± 103.1 Å$^2$ |
| Z-score | −1.2 |

formed by residues E62-A65. The C-terminal region is significantly populated with disorder promoting residues (76%)[50]. Apart from the limited chemical shift dispersion (Fig. 1b and Supplementary Fig. S4a), low relaxation rates and negative steady-state heteronulcear {$^1$H} - $^{15}$N NOEs of the residues in the C-terminal region (Fig. 1c), the absence of long-range NOEs are also indicative of the flexible nature of this region. During analysis, it was found that several residues (A65, L66, A67, L75) show conformational heterogenity which could play an importatnt role in neutralizing the toxin (Fig. 1b). Multiple conformations of the protein and high mobility of the disordered region are also seen in CcdA[49]. Structures of the MazEF4, EF7 and EF9 show that the C-terminal domains of the respective antitoxins are highly ordered in the TA complexes (Fig. 3c). It is expected that the C-terminal domain of MazE6 would become ordered when it neutralizes MazF6.

Autoregulation of *mazEF6* operon via feedback inhibition can occur by binding of either MazE6 or by MazEF6 TA complex, wherein the latter exerts a more stringent control[37,51]. Given the pathological significance of *Mtb* and the role of TA systems in the pathology and virulence of the organism, the sequences upstream of TSS of 2388 annotated coding genes of *Mtb* were further examined by Cortes et al., to determine consensus operator sequences, if any[43]. They have shown that 73% of the *Mtb* coding genes possess a conserved TANNNT -10 motif centered 7–12 bp upstream of the TSS. In this study, we have unequivocally established that the −10 upstream consensus sequence is TATACT and constitutes the operator site for *Mtb* MazEF6 expression and regulation. Our experimental data points to this sequence being the primary operator site, given the observation that the nMazE6 dimer binds to this operator DNA in 1:1 stoichiometry with nanomolar affinity (≈217 nM). Similarly, in the *E.coli* MazEF system, autoregulation occurs via binding of MazE or MazEF complex to the operator site at −10 motif except that the operator region in E. coli is palindromic in sequence[33,52].

The specificity of the interaction lies in the sequence of the protein as well as the operator DNA. The MazE proteins in *Mtb* are structurally conserved. Despite this structural conservation, the residues involved in DNA binding and making specific base contacts (residues in β-strand) are not conserved (Fig. 3f). Binding of MazE6 and nMazE6 to cognate operator DNA involves the same residues in the RHH motif (Fig. 5 and

Supplementary Fig. S8). In the case of MazE6, our data has shown that C-terminal domain does not interact with the DNA. The residues important for protein-DNA interaction are those present in the β-sheet and S28-S30 in helix α2. Similar results were observed by *Zhao et al.*, where they found that S25A and R26A mutations abolish DNA binding[37]. Structure of AtaR-AtaT-DNA complex support a similar method of interaction where residues in β-strand provide specificty and residues at the start of α2-helix anchor the protein on the DNA[53]. Specificity of interaction lies in the operator DNA sequence as well. For instance, the operator DNA for *mazEF6* (TATACT) and for *mazEF9* (TAGCAT) differ significantly in the type of nucleotide (purine or pyrimidine) present at the positions NNN in the consensus TANNNT sequence. Apart from the differences in the TANNNT sequence, *mazEF6* operator DNA has an additional GGT sequence upstream of TANNNT, while *mazEF9* operator DNA has TGG. These additional sequences are known to influence the promoter activity to different strength[43]. Mutation of the NNN nucleotide in the cognate DNA results in loss of interaction. This strongly suggests that the -10 promoter region is the high affinity and specific site and might be the nucleation site for *mazEF6* transcription autoregulation. These results are similar to those found for *ccdAB*[49] and *mazEF* system[33] from *E.coli* where the RHH domain of antitoxin CcdA and AbrB domain of MazE interacts with a high affinity operator site bearing the -10 promoter region.

A good agreement between the interaction studies carried out by ITC and NMR titration is observed (Table 2). A very small heat change was observed in ITC for interaction with non-cognate *mazEF9* operator DNA, while NMR titration studies showed very fast exchange without any line broadening. The observations from the ITC titrations are strongly supported by detailed lineshape analysis carried out using the modified Bloch-McConnell equations in TITAN. In the case of the interaction of nMazE6 with *mazEF9* operator the $k_{on}$ is almost 18 times slower while $k_{off}$ is ~14 times faster resulting in a significantly shorter life time for the complex as reflected in the much higher $K_d$. These studies have shown that MazE6 interacts with its operator DNA to regulate its transcription and is unlikely to interact with non-cognate *mazEF9* operator in-vivo.

Our studies have shown that in MazE6, the N-terminal DNA binding domain is highly structured, whereas the C-terminal Toxin binding domain is *intrinsically disordered*. The studies have shown that MazE6 and nMazE6 bind to the cognate operon with high affinity. The high resolution NMR studies have identified the key residues neccessary for this binding. Furthermore, we have shown that the MazE6 does not bind to the *mazEF9* operator DNA. Hence, it is safe to say that MazE6 is unlikely to regulate the transcription of *mazEF9* operon. The weak interaction between nMazE6 and *mazEF9* operator DNA in in vitro condition might be a reflection of the non-specific electrostatic interactions between the backbone of the non-cognate DNA and the RHH domain of nMazE6[54]. *Mtb* is one of the most lethal pathogenic bacteria which claims nearly 2 million lives per year worldwide[55]. The MazEF TA systems in *Mtb* are important for establishing the "Non Replicative Peristent" state. It has been observed that 70% of *Mtb*'s total RNA is susceptible to MazF6 RNase activity[56]. Mechanisms that hinder the MazE6-operator interaction will potentially disrupt the repression of transcription leading to increased levels of toxin activity as an 'intracellular death factor'. Synthetic oligonucleotides or other ligands that will compete for the DNA recognition motif or ligands that bind to other regions of MazE6 which will change its ability to interact with operator DNA can be designed based on the above study. The structure and dynamics of MazE6 and its' interaction with the operator DNA described above improves our understanding

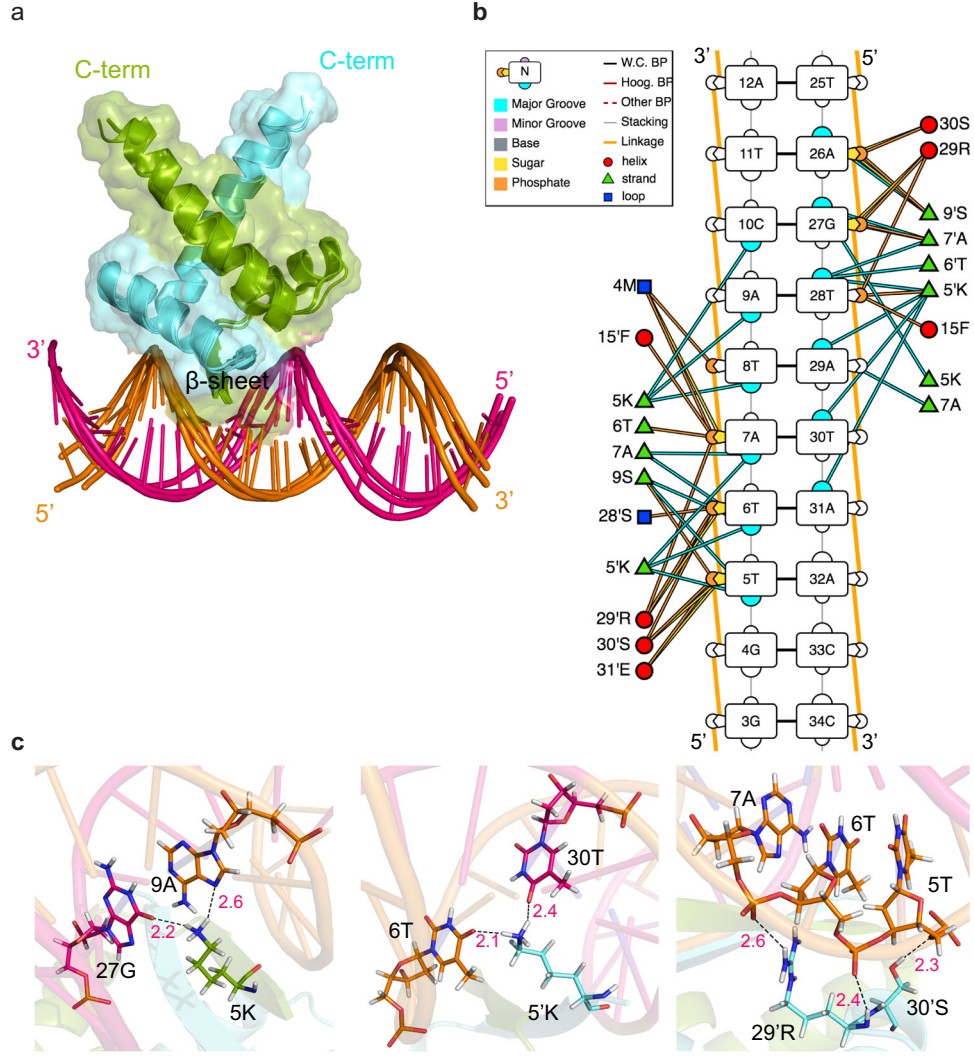

**Fig. 7 HADDOCK model of nMazE6-cognate operator DNA complex. a** Cartoon representation of the ensemble of the top three HADDOCK structures with the lowest interaction energies and lowest restraints violations. **b** Diagrammatic illustration of protein-DNA interactions. The sequence of the operator DNA is labeled in 5′–3′ directionality. Plausible interactions of protein residues with sugar, phosphate and major groove are represented by yellow, orange and blue colored solid lines. **c** Specific recognition of bases by residues in β-sheet and electrostatic interaction of residues from α2 helix with sugar-phosphate backbone of TATACT -10 motif in operator region. See text for details.

of the structural basis for this regulation and will aid in the rational design of inhibitors against *Mtb*.

## Materials and methods

**Protein expression and purification.** MazE6 protein was co-expressed with hexa-histidine tagged MazF6 (Toxin) from pETDuet-1 vector from MCS2 and MCS1 respectively using E. coli BL21(DE3) overexpression strains. Cells were grown at 37 °C with 1% primary inoculum. Induction of protein expression was carried out by addition of IPTG to a final concentration of 0.5 mM to the culture broth when the culture reached an O.D$_{600}$ 0.8. Cells were harvested after 16 h of induction at 20 °C by centrifugation (30 min/6000 *rpm*/4 °C) and resuspended in 50 mM Tris, 500 mM NaCl, 20 mM Imidazole, pH 8.0. The suspension was again centrifuged (30 min/13,000 *rpm*/4 °C) after lysing the cell by French Press at 1200 psi. Cell lysate containing MazEF6 complex was loaded on Ni-NTA affinity column. Denaturation buffer (50 mM Tris, 500 mM NaCl, 4M Guanidine Hydrochloride, pH 8.0) was then passed to separate MazF6 from MazE6. MazE6 was collected in flowthrough during denaturaion step. The bound MazF6 protein was eluted with a gradient of 0–1 M Imidazole. MazE6 was refolded by dialysing the protein in buffer comprising of 50 mM Tris, 300 mM NaCl, 50 mM Arginine Hydrochloride and 10% Glycerol at pH 8.0. Final gel filtration step was performed against 20 mM phosphate buffer containing 20 mM NaCl, 1 mM EDTA, pH 7.0.

For nMazE6, the pET-21a(+)-*nmazE6* vector with a hexa-histidine tag and TEV cleavage site at the N-terminus was transformed into E. coli BL21(DE3) overexpression strains. Cells were grown at 37 °C with 1% primary inoculum. Induction of protein expression was carried out by addition of IPTG to a final

concentration of 0.5 mM to the culture broth when the culture reached an O.D$_{600}$ 0.8. Cells were harvested after 16 h of induction at 20 °C by centrifugation (30 min/6000 *rpm*/4 °C) and resuspended in 50 mM Tris, 500 mM NaCl, 20 mM Imidazole, pH 8.0. The suspension was again centrifuged (30 min/13,000 *rpm*/4 °C) after lysing the cell by French Press at 1200 psi. Purification was carried out using Ni-NTA affinity chromatography. Purified fusion protein was then dialyzed into TEV cleavage buffer (50 mM Tris, 250 mM NaCl, pH 8.0) and digested with TEV protease. The reaction products were separated by Ni-NTA affinity chromatography. Final step of gel filtration was performed against 20 mM Potassium Phosphate, 20 mM NaCl, 1 mM EDTA, pH 7.0.

For MazE6c, the pET-21a(+)-*mazE6c* vector with a hexa-histidine tag followed by Cytochrome $b_5$ tag at the N-terminus was transformed into E. coli BL21(DE3) overexpression strains. Cells were grown at 37 °C with 1% primary inoculum. Induction of protein expression was carried out by addition of IPTG to a final concentration of 0.5 mM to the culture broth when the culture reached an O.D$_{600}$ 0.8. Cells were harvested after 5 h of induction at 37 °C by centrifugation (30 min/6000 *rpm*/4 °C) and resuspended in 50 mM Tris, 500 mM NaCl, 20 mM Imidazole, pH 8.0. Purification was carried out using Ni-NTA affinity chromatography. Purified fusion protein was then dialyzed into thrombin cleavage buffer (50 mM Tris, 100 mM NaCl, 200 mM CaCl$_2$, pH 8.0) and digested with thrombin (Sigma Chemicals Ltd.). The reaction products were separated by gel filtration against appropriate NMR sample buffer.

Stable isotope precursors for protein production were purchased from Sigma Chemicals or from Cambridge Isotope Laboratories. Protein samples that were isotopically enriched in $^{15}$N or $^{13}$C,$^{15}$N were prepared by culturing cells bearing the appropriate plasmids in minimal media containing $^{15}$NH$_4$Cl or $^{15}$NH$_4$Cl and $^{13}$C$_6$

- glucose as the sole sources of nitrogen and carbon dissolved in $H_2O$[57]. Protein samples that were enriched in $^2H$, $^{13}C$ and $^{15}N$ were prepared by culturing cells in minimal media containing $^2H_7, ^{13}C_6$ - glucose, and $^{15}NH_4Cl$ as carbon and nitrogen sources respectively, dissolved in 99% enriched $D_2O$. ILV-methyl protonated samples were prepared using $^2H_7, ^{13}C_6$ - glucose and $^{15}NH_4Cl$ as carbon and nitrogen sources respectively, supplemented with $3,3-^2H_2$, $^{13}C_4$ - α-ketobutyric acid and $3-^2H, ^{13}C_5$ - α-ketoisovaleric acid[58], in 99% $D_2O$ as solvent.

**NMR sample preparation**. NMR samples of MazE6, nMazE6 and MazE6c were prepared by dissolving protein in 90% aqueous buffer (20 mM KPi pH 7.0, 50 mM NaCl, 1 mM EDTA) and 10% $D_2O$, except for the 3D HCCHTOCSY(100% $D_2O$). The concentration of proteins for all experiments ranged from 300 to 600 μM except for titraton experiments, where the initial protein concentraion was set to 100 μM. Chemical shifts were referenced to external DSS.

**NMR data acquisition**. All spectra for assignment and structure calculation were acquired on an Agilent (Varian) DDS2 600 MHz spectrometer equipped with a cryogenically cooled cold-probe fitted with a pulsed field gradient (z-axis only) accessory at 30 °C for MazE6 and at 25 °C for nMazE6 and MazE6c.

**NMR data processing and analysis**. NMR data were processed on Intel work-stations running on CentOS 7 operating systems using NMRPipe/NMRDraw processing software[59]. Spectra were analyzed using the CCPNMR (versions 2.1–2.4) suite of programs $^{ccpn}$.Backbone atoms of the proteins were assigned using HNCACB, CBCA(CO)NH (HN(CO)CACB for MazE), HNCO and HN(CA)CO. Aliphatic side-chains were assigned using correlations from C(CO)NH and HC(CO)NH . For aromatic side chains, $^1H$-$^1H$ TOCSY, $^1H$-$^1H$ NOESY, $^{13}C$-HSQC and $^{13}C$-NOESY-HSQC spectra were used.

**nMazE6 structure calculation**. Structures of nMazE6 were calculated by CYANA version 2.1[60] using experimentally determined distance and dihedral angle restraints. Unambiguously assigned NOE correlations from $^{13}C$-NOESY-HSQC, $^{15}N$-NOESY-HSQC, $^1H$-$^1H$ NOESY spectra were used to derive distance restraints from their peak intensities. A lower bound of 1.8 Å was set for all distance restraints. To generate hydogen bond restraints, knowledge of secondary structure was used. Upper distance bounds of 2.2 Å for NH ⋯ O bonds and 3.2 Å for N ⋯ O bonds were set as restraints between hydrogen bonded donor and acceptor pairs. Backbone dihedral angles were computed by DANGLE, a program embedded within CCPNMR[61], from $^{13}C^\alpha$, $^{13}C^\beta$ and $^{13}C'$ secondary CSI. During structure calculation, 999 random conformers were subjected to 10,000 steps of annealing to obtain an ensemble of 50 structures of acceptable stereochemical quality. Final structural refinement with water was done by XPLOR-NIH[62] with output files from CYANA. First, CYANA structures were converted into the XPLOR format using pdbstat[63] with a 10% range for the upper bound (upper limit for NOEs), and the van Der Waals radius was used as the lower bound (lower limit for NOEs). The protein structure validation suite wwPDB Validation server[64] has been used to check the quality of the structure. The twenty lowest energy structures were used for further analysis.

**MazE6-operator binding study by Isothermal Titration Calorimetry (ITC)**. A stretch of 18 bp upstream of *mazEF6* TSS which invloves −10 promoter region was taken for interaction study named as *mazEF6* operator DNA. ITC experiments were carried out using MicroCal VP-ITC instrument (GE) at 303 K. Samples were degassed before the experiment. For the titrations, 10–30 μM of the DNA was used in sample cell and was titrated with 100–300 μM of nMazE6 respectively. Thirty injections of the titrant were performed at an interval of 180 s. Interaction studies of nMazE6 with *mazEF6* mutated operator DNA and *mazEF9* operator DNA were done at a concentration of 10 μM DNA in sample cell and 100 μM nMazE6 in syringe. A control experiment was done with scrambled DNA having same nucleotide base pairs as the cognate DNA but with scrambled sequnce. All the DNA sequences used in the study are listed in Table S1. For heats of dilution, separate protein-in-buffer experiments were performed and subtracted from the integrated heat and the data were fit for a one site-binding model. All the parameters were kept floating during data fitting. The experiments were repeated twice or thrice for consistency.

**Chemical shift mapping of MazE6-operator binding**. Binding of antitoxin to its cognate *mazEF6* operator DNA, *mazEF6* mutated operator DNA and non-cognate *mazEF9* operator DNA was also probed by chemical shift mapping of MazE6 and nMazE6 upon addition of DNA. Either 50 or 100 μM $^{15}N$- labeled protein (MazE6 or nMazE6) was titrated with unlabeled DNA ranging from 0 to 100 or 200 μM at 298 K.

To determine the combined and weighted backbone amide chemical shift perturbation (Δδ) was calculated using the standard formula:

$$\Delta\delta = [(\Delta\delta_H)^2 + (\Delta\delta_N/5)^2]^{1/2} \tag{1}$$

where $\Delta\delta_H$ and $\Delta\delta_N$ are the chemical shift changes (in ppm) at the beginning and the end of the titration.

**NMR titration fitting by TITAN**. Two dimensional lineshape analysis of NMR titration data was done by TITAN software package. It numerically simulates the evolution of magnetization during a pulse sequence in the presence of chemical exchange, in the Liouville space. All the $^1H$- $^{15}N$- HSQC spectra of a titration series were processed identically. Two-state binding model was used to fit the following twenty-three residues of nMazE6 showing perturbation in the chemical shift: 4M, 5K, 7A, 12D, 13E, 14T, 15F, 17R, 18V, 19S, 20R, 22A, 23S, 25L, 26G, 27M, 32F, 36A, 39R, 41L, 42H, 44L and 46A.

**$^{15}N$-backbone dynamics**. A series of 2D $^1H$-$^{15}N$ HSQC spectra with different $T_1$ and $T_{1\rho}$ relaxation delay time intervals were acquired to measure spin-lattice relaxation rate $R_1$ ($1/T_1$), and spin-spin relaxation rate $R_{1\rho}$ ($1/T_{1\rho}$) respectively. Each $t_1$ data time point was signal averaged over 32 transients. The pulse schemes described by Farrow et al. was used to acquire the data sets[65]. For measuring $T_1$, relaxation delays were set to 10, 50, 100, 200, 400, 600, 800, 1000, 1100, 1300, and 1500 ms. For $T_{1\rho}$ the relaxation delays were set to 2, 5, 10, 20, 30, 40, 50, 60, 70, and 80 ms. $^1H$-$^{15}N$ Heteronuclear NOEs were measured from the ratio of peak intensities ($I_{on}/I_{off}$) with and without the saturation of the amide protons for 3 s.

**Structure calculation of nMazE6-operator complex**. Data driven structural model of the complex was generated by HADDOCK 2.4. The ensemble of 20 lowest energy structure of nMazE6 was used for docking on the energy minimized 3D model of operator DNA generated by Web 3DNA (w3DNA) 2.0 server[66]. The residues showing Chemical Shift Perturbation (CSP) more than the average perturbation were investigated for their location in the structure and accessiblity. The active residues defined in protein for docking were: 5K, 7A, 9S, 15F, 29R, 30S, 5K′, 7A′, 9S′, 15F′, 29R′, 30S′. For DNA, 'all nucleotides' were considered active in initial runs. The nucleotides to which the initial runs pointed to, were taken as active residues in final run : 5–12 and 25–32. The docked structures with the lowest intermolecular energies given as output from HADDOCK were analyzed using PyMOL[67].

**Statistics and reproducibility**. All ITC experiments described were performed in independent duplicates. NMR titration for nMazE6-cognate operator DNA was performed in independent duplicates.

**Reporting summary**. Further information on research design is available in the Nature Research Reporting Summary linked to this article.

## Data availability
The datasets generated during and/or analyzed during the current study are available from the corresponding author on reasonable request. The $^1H$, $^{13}C$ and $^{15}N$ assigned resonances of MazE6 and nMazE6 have been deposited in the BioMagResBank (http://www.bmrb.wisc.edu/) under accession number 51226 and 51227 respectively. The structural coordinates and experimentally derived restraints for nMazE6 and nMazE6-operator DNA complex have been deposited in the Protein Data Bank with accession number 7WJ0 and 7WNR respectively. Source data are provided with this paper (Supplementary Data 1).

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

## Acknowledgements

Authors acknowledge the Department of Science and Technology (DST) and Department of Biotechnology (DBT), India, for the NMR facility in Molecular Biophysics Unit at the Indian Institute of Science (IISc), Bangalore. The authors acknowledge support for NMR instrument maintenance from the IISc-IOE funds and DBT-IISc partnership program for the Mass-Spectrometry . The authors thank Dr. Ashok Sekhar, MBU for useful discussions.

## Author contributions

Project was concieved by S.P.S. and K.K. NMR data acquisition and analysis was done by K.K. and S.P.S. ITC data was acquired by K.K. and analyzed by K.K. and S.P.S. HAD-DOCK runs were performed by K.K. and analyzed by K.K. and S.P.S.

## Competing interests

The authors declare no competing interests.
