## [Peer Review File · Communications Biology]

Reviewers' comments:

Reviewer #1 (Remarks to the Author):

The manuscript characterized the MazEF6 TA system from *Mycobacterium tuberculosis* biochemically and structurally in order to gain knowledge of the molecular recognition of the DNA promoter and the autoregulation mechanism of the MazEF6 system. The authors obtained the NMR structure of the N-terminal domain of the antitoxin (nMazE6) and studied its interactions with the DNA promoter. Via ITC titration, nMazE6 was found to associate with the DNA promoter fragment with a high affinity. On the other hand, the C-terminal region was intrinsically disordered. Overall, the manuscript mainly focuses on the structural aspects of the TA system, which does not provide much insight into the design of chemotherapeutic agents. Therefore, the manuscript should be sent to a more specialized journal. In addition, the info obtained from the structural studies did not convey any new messages other than the ones we already knew from prior studies in the field, thus did not provide any conceptual advancements.

Miscellaneous problems:

1. There are a number of Mtb MazEF systems that have been well characterized in the literature. Why the authors chose MazEF6 to study and its significance was unclear;
2. The authors only studied the biochemistry and structure of the antitoxin but left the toxin out completely. How would the toxin stabilize the antitoxin and how would the TA complex associate with the DNA promoter?
3. The authors obtained some structural information from the apo-nMazE6 protein and nMazE6-DNA complex. How different were they from other well-studied Mtb MazEF systems and what is the most salient feature of MazEF6?
4. ITC titration: the heat generated in figure 9A was too little.
5. At numerous occasions, the authors forgot to insert the figure number into the text, and phrases like "see Figure ??B" were seen quite often. pET-21a(+) was spelled as "peT21(a)" as well.

Reviewer #2 (Remarks to the Author):

In the manuscript by Kumari and Sarma the authors use primarily NMR spectroscopy, but also other biophysical techniques, to determine the structure and DNA binding of the antitoxin protein MazE6. The description of the experiments and results are clearly presented and analyzed, and the conclusions are supported by the data. The atomic information presented by the authors represents an important contribution to our understanding of the type II toxin-antitoxin mechanism.

There are no major concerns regarding the text and figures, but there are several smaller points that should be addressed:

- in the introduction paragraph 6, it would be useful to include a simple schematic (perhaps within Figure 1 as a new panel A) to show how this type II toxin-antitoxin system works (somewhat along the lines of the current Figure 11B). This would help to better understand the subsequent experiments for those that are not experts in this system

- on page 6 it is mentioned that the 15N-HSQC spectra for full-length MazE6 and nMazE6 are similar for the disperse amide crosspeaks, however, in Figure S4 the spectra are actually fairly different. There are peaks that likely correspond, but the spectra are otherwise moderately perturbed upon truncation of the C-terminal region. Since the chemical shifts of both dimers are assigned, can the authors more precisely report on the main areas of chemical shift perturbation between the full-length and nMazE6?

- In table 1, the structure determination statistics, it is now clear if the numbers in the table represent

the total number of restraints used or are just one-half (i.e. just for one monomer). The fact that there are an odd number of inter-subunit restraints makes it likely that these are just one-half of the restraints? Maybe the authors could add in parentheses the actual total number of restraints used for the dimeric complex for clarity?

- In figure 5, the orientation of the structure is the same for panels B and D, but different for panel A. Is there a reason for this? If not, it might be clearer to also have panel A in the same orientation

- on Page 10 the rates should also have the uncertainty reported (i.e. +/- standard deviation). Similarly it should be reported if the ITC was done in replicate, and again there should be the uncertainty reported

- for the HADDOCK docking it was noted that a subset of DNA bases were used for the docking - was this region already based on some previously docking attempts? Otherwise it seems that this selection decision would bias a bit the final outcome. Perhaps the authors could expand a bit on how these nucleotides were chosen?

- I think a very interesting aspect of the MazE6 dimer-DNA interaction is that although the MazE6 is symmetric it binds in a non-symmetric way to the DNA. Can the authors provide some rationale as to how or why this can occur?

- The model and strategy outlined in Figure 11 is perhaps outside of the context of the current manuscript and could be removed. It is largely speculative, and does not stem directly from the reported data in this manuscript. Instead a few sentences could be retained to indicate that improved knowledge of the molecular details can help to design future strategies.

There are also some small text suggestions:

- pET21 is often written as pET21

- the heteronuclear NOE NMR setup should be moved to the ¹⁵N backbone dynamics section in the materials in methods

- a few of the supplemental figures are listed as '??' in the text

Reviewer #3 (Remarks to the Author):

The mazEF system is a prominent toxin-antitoxin system with homologs in Gram positive and Gram negative bacteria. MazF toxins act as RNases and MazE antitoxins regulate MazF activity as well as transcription of the mazEF operon. The manuscript analyses the structural basis of the interaction between MazE6 of *M. tuberculosis* and its cognate operator sequence. The proteins MazE6, a truncated version of MazE6, and MazF6 were purified and used for CD, NMR and ITC and further structural biology techniques. The study analyses and quantifies the interaction between MazE6 and its cognate DNA binding site and proposes a structure of the protein-DNA complex by a docking simulation.

Remarks:

Abstract:

- The author's should highlight, why they chose mazEF6 (instead of other MazEF paralogs of MazEF in *M. tuberculosis*) for their investigations

- 1st sentence, please clarify that the MazEF TA systems are not exclusive to *M. tuberculosis*

- At the end, I suggest to remove "unequivocally", to keep the description neutral

Introduction:

- Not all TA systems are encoded as bicistronic operons (e.g. type I TA systems), please clarify
- Write *Mycobacterium tuberculosis* in full before using the abbreviation Mtb
- Briefly explain the genetic set-up and mode of control of type II TA systems
- Beyond the MazEF systems of Mtb, structural data are also available for other homologs, e.g. in *E. coli* (DOI 10.1074/jbc.M116.715912) or *S. aureus* (doi: 10.1093/nar/gku266)
- It would be helpful to state the number of residues of MazE6 and to provide more information on how many C-terminal amino acids are lacking in nMazE6

Materials and Methods:

- Is it recommended to provide temperature as K instead of °C?

Results:

- Figure 1A may be too special for the readership of *Communications Biology*. Please consider shifting it to supp. material. Same for parts of Fig. 2, Fig. 4, as well as Fig. 8
- correct question marks in figure designations three times (and also in the discussion)

Discussion:

- Fig. 11 A: Consider labeling the sequences of the alignment with the MazE type and instead write the PDB accession numbers into the figure legend

Miscellaneous:

- There is a number of typos in the text and also in the References (e.g. when umlauts are used), please check carefully

Manuscript ID - COMMSBIO-22-1497-T
Reponse to Reviewers

Khushboo Kumari and Siddhartha P Sarma

Response to Reviewer 1

The manuscript characterized the MazEF6 TA system from *Mycobacterium tuberculosis* biochemically and structurally in order to gain knowledge of the molecular reception of the DNA promoter and the autoregulation mechanism of the MazEF6 system. The authors obtained the NMR structure of the N-terminal domain of the antitoxin (nMazE6) and studied its interactions with the DNA promoter. Via ITC titration, nMazE6 was found to associate with the DNA promoter fragment with a high affinity. On the other hand, the C-terminal region was intrinsically disordered. Overall, the manuscript mainly focuses on the structural aspects of the TA system, which does not provide much insight into the design of chemotherapeutic agents. Therefore, the manuscript should be sent to a more specialized journal. In addition, the info obtained from the structural studies did not convey any new messages other than the ones we already new from prior studies in the field, thus did not provide any conceptual advancements.

We thank the reviewer for his/her critical comments and observations. The study presented by us provides insights into the structure , kinetics and thermodynamics of the association of the MazE6 protein with it's operator DNA. MazE6 is an important protein because (i) it acts as the antidote of the toxin (MazF6) which plays an important part in the establishment of Non-replicating persistent state of *Mycobacterium tuberculosis* and (ii) it also regulates the transcription of its operon. Furthermore, the study has established the high affinity operator sequence and it has been verified through well-designed mutational analysis.

Query 1

There are a number of Mtb MazEF systems that have been well characterized in the literature. Why the authors chose MazEF6 to study and its significance was unclear.

Mtb must adapt to nutrient-limiting, hypoxic and other adverse conditions to remain dormant and persist in the host for which the TA systems are critical. Tiwari *et. al.* (2015) have shown that overexpression of MazF3, MazF6 and MazF9 reduced bacterial loads by 8.0-, 22.0- and 18.0-fold, respectively. While, the overexpression of the remaining MazF homologues, MazF1, MazF2, MazF4, MazF5, MazF7 and MazF8 did not inhibit *M. bovis* BCG growth. Further, the expression level of *mazF6* transcripts have been shown to be even higher than those of *mazF3* and *mazF9* under nitrosative, hypoxic and nutrient stress conditions and upon drug treatment [1]. Thus, the MazEF6 TA system plays a critical role in the establishment of the Non Replicative Persistent (NRP) state of *Mtb*.

The structural details for MazEF4, MazEF7 and MazEF9 complexes are known. However, no data is available for their interaction with their cognate operator DNA. Furthermore, the structure of the toxin component. i.e. MazF6 with the substrate DNA is known. Here, we have studied the structure, kinetics and thermodynamics of the interaction of MazE6 with it's operator DNA, thus establishing it's role as a transcriptional regulator, a first for this class of proteins in *Mtb*. This study thus complements the earlier studies of the MazEF6 TA systems and thus enables extrapolation of the these interactions to

other homologous MazEF systems.

The manuscript has been revised to indicate the importance of the MazEF6 TA system in the Introduction (paragraph 6) as well as Abstract sections.

Query 2

The authors only studied the biochemistry and structure of the antitoxin but left the toxin out completely. How would the toxin stabilize the antitoxin and how would the TA complex associate with the DNA promoter?

Studies by Zhao *et. al.* [2] and by Gopinath *et. al.* [3], clearly established that the toxin (MazF6) alone does not bind to the operator. Furthermore Gopinath, *et. al.* have shown that toxin and antitoxin form higher order oligomers, *viz.*, hetero-hexamers and hetero-octamer complexes and that these higher order oligomers also bind to the DNA.

Attempts on our part to structurally characterize these higher order MazEF6 protein complexes or as ternary complexes with operator DNA, by X-ray diffraction were not successful because of the heterogeneity inherent in the system. In effect, these MazEF6 protein oligomers range in size from 92 kDa (TA complex) to 120-150 kDa (TA-operator DNA ternary complex) which are too large to study by NMR. Future experiments have been designed to study the ternary complex by Cryo-Electronmicroscopy.

Query 3

The authors obtained some structural information from the apo-nMazE6 protein and nMazE6-DNA complex. How different were they from other well-studied Mtb MazEF systems and what is the most salient feature of MazEF6?

The multiplicity of the MazEF TA systems in *Mtb.*, strongly suggest that they are likely to have similar tertiary structures. However, given the uniqueness of each of the MazEF TA systems in cellular homeostasis once again suggests that the details of the interactions with either substrate or operator-DNA are critically dependent on local structure and conformation. As we have shown, the MazE6 antitoxin does not interact with the MazE9 operator DNA and the same is possibly true with the cognate operator DNA's of other MazEF systems in *Mtb.* The devil does indeed lie in the details.

The figure below shows the structural alignment of MazE6 with other structural homologs (Fig. 1C, F). While, the structural scaffold remains the same *i.e.*, RHH, there are few differences that makes every protein unique. The residues in beta strand which are responsible for binding the operator DNA are significantly different. 5Lys and 9Ser that are responsible for making specific base contact with the operator DNA are not conserved (Fig. 1F)

One unique feature found in the structure of MazE6 is that the N-terminal domain is stabilised by a network of aromatic residues (Fig. 1E). While, MazE4, MazE7 and MazE9 have two, one and two aromatic residues in their N-terminal domain respectively, MazE6 has four of them. Four aromatic residues from each monomer make a donut like aromatic cluster (Fig. 1E) which stabilises the protein and makes a sort of hydrophobic core of the protein. Such a core is not found in any of the structures solved.

These points are highlighted in manuscript paragraph 2, 3, 4 and 5 of Discussion.

Query 4

ITC titration: the heat generated in figure 9A was too little.

The heat change in the reaction is small because the concentrations used were: 10uM DNA (titrate) and 100uM nMazE6 (titrant). ITC titrations at higher concentrations (30uM and 300uM respectively) were

Figure 1: **(A)** Structural alignment of nMazE6 with other MazE proteins showing the conservation of DNA binding motif (RHH) in *Mtb* MazE proteins and *Bacillus subtilis* MazE. The residues (present in beta-strand) making specific base contacts with DNA are significantly different leading to difference in specificity. s001A, 6a6xC, 5xe3E, 6kytC and 4me7F represents nMazE6, MazE7, MazE4, MazE9 and *Bacillus subtilis* MazE respectively. Structural alignment was done using the DALI server[4] **(B)** Plausible strategy for artificial activation of MazEF6 TA system 1. Based on the complex structure, design of mechanism/factor/drug to disrupt MazE6-operator interaction 2. Introduction of the factor/drug responsible for inhibition of MazE6-operator interaction either by binding to nMazE6 or operator DNA 3. Loss of auto-repression of mazEF6 operon resulting in uncontrolled expression of MazEF6 complex 4. Removal of the factor/drug leading to tight repression of the mazEF6 operon 5. Introduction of Lon protease resulting in degradation of MazE6 and activation of MazF6

also carried out and produced higher heat change shown in Figure 2 below (not included in manuscript) but yielding identical K_d and other thermodynamic parameters. The change in enthalpy for the reaction is significant. It is important to note that the thermodynamic parameters derived from ITC studies agree very well with those derived from NMR titration studies (50uM nMazE6, 5-100uM DNA titrant) via a complete 2D Line-shape analysis and fitting the Bloch -McConnell equations (Manuscript Table 2).

Figure 2: **Isothermal titration calorimetry on nMazE6-DNA binding** Isothermogram of the titration of operator DNA with nMazE6 at 30° C at different concentrations.

Query 5

At numerous occasions, the authors forgot to insert the figure number into the text, and phrases like see Figure ??B were seen quite often. pET-21a(+) was spelled as peT21(a) as well.

We thank the reviewer for pointing this out. The figure references have been corrected and pet21(a) has been now changed to pET-21a(+)

Response to Reviewer 2

In the manuscript by Kumari and Sarma the authors use primarily NMR spectroscopy, but also other biophysical techniques, to determine the structure and DNA binding of the antitoxin protein MazE6. The description of the experiments and results are clearly presented and analyzed, and the conclusions are supported by the data. The atomic information presented by the authors represents an important contribution to our understanding of the type II toxin-antitoxin mechanism.

We thank the reviewer for his/her critical analyses of the data and results and valuable comments.

Query 1

In the introduction paragraph 6, it would be useful to include a simple schematic (perhaps within Figure 1 as a new panel A) to show how this type II toxin-antitoxin system works (somewhat along the lines of the current Figure 11B). This would help to better understand the subsequent experiments for those that are not experts in this system

We thank the reviewer for this suggestion and we have included the working mechanism in paragraph-3 of the introduction section. We have also included a figure that summarizes this mechanism in Figure 1. All the figures are provided at the end of this document.

Query 2

On page 6 it is mentioned that the ^{15}N -HSQC spectra for full-length MazE6 and nMazE6 are similar for the disperse amide crosspeaks, however, in Figure S4 the spectra are actually fairly different. There are peaks that likely correspond, but the spectra are otherwise moderately perturbed upon truncation of the C-terminal region. Since the chemical shifts of both dimers are assigned, can the authors more precisely report on the main areas of chemical shift perturbation between the full-length and nMazE6?

Differences in the chemical shift are observed for the following reasons:

(A) Full-length MazE6 was overexpressed and purified without any tags. The clone used for expression and purification of nMazE6 required a N-terminal His tag for optimal expression and purification. The tag was removed via TEV cleavage leaving behind three additional residues as a consequence of cloning exigencies. This results in apparent differences in the HSQC spectra of the two proteins.

(B) In addition, absence of the C-terminal domain causes changes in the spectrum in the C-terminal region of the nMazE6. Residues such as 30S, 35K, 40Y, 43E which are present in the C-terminal region in nMazE6 were not observed in full-length MazE6 presumably due to motional or exchange broadening. Removal of C-terminal domain has had manifest effects on conformational exchange and hence the correlations in the HSQC are clearly visible in nMazE6. It also caused shift in the position of few residues which are terminal in position (42H, 48L, 49L, 50T).

The reasons for the difference in shifts have been mentioned in the legend of the figure (Now Figure S6).

Query 3

In table 1, the structure determination statistics, it is now clear if the numbers in the table represent the total number of restraints used or are just one-half (i.e. just for one monomer). The fact that there are an odd number of inter-subunit restraints makes it likely that these are just one-half of the restraints? Maybe the authors could add in parentheses the actual total number of restraints used for the dimeric complex for clarity?

In Table 1, the numbers represent the **total number of restraints** used for the structure calculation of the dimer. Restraints for each sub-unit were doubled and symmetric restraints were introduced for NOEs observed across the interface. This information is now included in the Table 1 footnote.

We had made an error in the enumeration of the total number of inter-subunit restraints and therefore, it was submitted as an odd number. The error has been rectified.

Query 4

In figure 5, the orientation of the structure is the same for panels B and D, but different for panle A. Is there a reason for this? If not, it might be clearer to also have panle A in the same orientation

(A) The orientation of the structure in figure A is chosen to emphasize the structural symmetry in the dimer.

(B) In panel B, the orientation is chosen to clearly display the important secondary structure elements in nMazE6 .

(C) Similarly, in Panel D, orientation is chosen to display the aromatic cluster that is important for the stability of the dimeric protein. If Panel B and D are oriented as Panel A, the secondary structure

elements and aromatic cluster will not be clearly visible. The figure has been suitably modified to indicate the rotation operations that result in the orientations in each panel. (Refer Figure 3 at the end of the document)

Query 5

on Page 10 the rates should also have the uncertainty reported (i.e. +/- standard deviation). Similarly it should be reported if the ITC was done in replicate, and again there should be the uncertainty reported

We have now incorporated the uncertainty in the rates as well as in ITC data.

Query 6

for the HADDOCK docking it was noted that a subset of DNA bases were used for the docking - was this region already based on some previously docking attempts? Otherwise it seems that this selection decision would bias a bit the final outcome. Perhaps the authors could expand a bit on how these nucleotides were chosen?

We thank the reviewer for his / her comment. To avoid bias, several docking runs were performed before arriving at the bases that were selected as “active residues” on DNA. In initial trials, all nucleotides were selected as “active residues” . These trials strongly indicated that nucleotides in the -10 region in this operator DNA sequence were strong candidates for interaction with the transcription repressor.

The results of the above described HADDOCK run is shown in the figure below (Not included in the manuscript). The final HADDOCK results yield better scores (Manuscript Table 3).

Figure 3: HADDOCK model of nMazE6-cognate operator DNA complex with all nucleotides considered as 'active residues'.

That the -10 region is the most likely candidate as a transcription factor binding site has also been suggested by Cortes *et. al.*, who found that 73% of the *Mtb* coding genes possess a conserved TANNNT -10 motif centered 7-12 bp upstream of the Transcription start site(TSS) [5].

To verify these results we had also carried interaction studies using DNA constructs harbouring mutations in the -10 region. The inability of nMazE6 to bind to the mutated DNA is a validation of the above results.

Query 7

I think a very interesting aspect of the MazE6 dimer-DNA interaction is that although the MazE6 is symmetric it binds in a non-symmetric way to the DNA. Can the authors provide some rationale as to how or why this can occur?

The operator regions present in the well studied *E. coli* operons (e.g., *lac*-operon, *trp*-operon [6] and *mazEF*-operon) have either palindromic or repeat or inverted repeat sequences to which symmetric molecules of regulatory protein interact in a symmetric manner. Whereas, in the case of *M.tb* operons, the operator sequences are generally devoid of palindromic/repeat sequences [5]. Therefore, the regulatory protein (nMazE6 in case of *mazEF6*-operon) binds asymmetrically to the operator region.

Query 8

The model and strategy outlined in Figure 11 is perhaps outside of the context of the current manuscript and could be removed. It is largely speculative, and does not stem directly from the reported data in this manuscript. Instead a few sentences could be retained to indicate that improved knowledge of the molecular details can help to design future strategies.

As per the suggestion, we have removed the model and strategy and added a new schematic at the start (Manuscript Fig. 1A) showing how Type II TA systems work.

Query 9

There are also some small text suggestions:

-pET21 is often written as pET21

-the heteronuclear NOE NMR setup should be moved to the ¹⁵N backbone dynamics section in the materials in methods

-a few of the supplemental figures are listed as '??' in the text

We thank the reviewer for pointing out the mistakes. There were several of them and it has been taken care of.

-pET21a(+) has been written in the correct form : pET-21a(+).

-the heteronuclear NOE NMR setup has been moved to dynamic section.

-The figure references has been corrected.

Response to Reviewer 3

The mazEF system is a prominent toxin-antitoxin system with homologs in Gram positive and Gram negative bacteria. MazF toxins act as RNases and MazE antitoxins regulate MazF activity as well as transcription of the mazEF operon. The manuscript analyses the structural basis of the interaction between MazE6 of *M. tuberculosis* and its cognate operator sequence. The proteins MazE6, a truncated version of MazE6, and MazF6 were purified and used for CD, NMR and ITC and further structural biology techniques. The study analyses and quantifies the interaction between MazE6 and its cognate DNA binding site and proposes a structure of the protein-DNA complex by a docking simulation.

We thank the reviewer for his/her critical analyses of the data and results and valuable comments.

Query 1,2,3 (Abstract)

- 1. The author's should highlight, why the chose mazEF6 (instead of other MazEF paralogs of MazEF in *M. tuberculosis*) for their investigations.
- 2. 1st sentence, please clarify that the MazEF TA systems are not exclusive to *M. tuberculosis*
- 3. At the end, I suggest to remove "unequivocally", to keep the description neutral

Mtb must adapt to nutrient-limiting, hypoxic and other adverse conditions to remain dormant and persist in the host for which the TA systems are critical. Tiwari *et. al.* (2015) have shown that overexpression of MazF3, MazF6 and MazF9 reduced bacterial loads by 8.0-, 22.0- and 18.0-fold, respectively. While, the overexpression of the remaining MazF homologues, MazF1, MazF2, MazF4, MazF5, MazF7 and MazF8 did not inhibit *M. bovis* BCG growth. Further, the expression level of *mazF6* transcripts have been shown to be even higher than those of *mazF3* and *mazF9* under nitrosative, hypoxic and nutrient stress conditions and upon drug treatment [1]. Thus, the MazEF6 TA system plays a critical role in the establishment of the Non Replicative Persistent (NRP) state of *Mtb*.

The structural details for MazEF4, MazEF7 and MazEF9 complexes are known. However, no data is available for their interaction with their cognate operator DNA. Furthermore, the structure of the toxin component. i.e. MazF6 with the substrate DNA is known. Here, we have studied the structure, kinetics and thermodynamics of the interaction of MazE6 with it's operator DNA, thus establishing it's role as a transcriptional regulator, a first for this class of proteins in *Mtb*. This study thus complements the earlier studies of the MazEF6 TA systems and thus enables extrapolation of the these interactions to other homologous MazEF systems.

The manuscript has been revised to indicate the importance of the MazEF6 TA system in the abstract as well as Introduction (paragraph 6).

Query 4-8 (Introduction)

- 4. Not all TA systems are encoded as bicistronic operons (e.g. type I TA systems), please clarify
- This sentence has been removed.

- 5. Write *Mycobacterium tuberculosis* in full before using the abbreviation *Mtb*

-Corrected

- 6. Briefly explain the genetic set-up and mode of control of type II TA systems

-The change has been introduced in the paragraph 3.

- 7. Beyond the MazEF systems of *Mtb*, structural data are also available for other homologs, e.g. in *E. coli* (DOI 10.1074/jbc.M116.715912) or *S. aureus* (doi: 10.1093/nar/gku266)

-Changes have been introduced and other structures have been referred too in paragraph 5.

- 8. It would be helpful to state the numer of residues of MazE6 and to provide more information on how many C-terminal amino acids are lacking in nMazE6

- The number of residues present in full length MazE6 (1-82) and nMazE6 (1-49) have been mentioned in the introduction .

Query 9 (Materials and Methods)

Is it recommended to provide temperature as K instead of C?

- Kelvin was used as it is the SI unit. For convenience, we have changed the unit to °C.

Query 10-11 (Results)

10. Figure 1A may be too special for the readership of *Communications Biology*. Please consider shifting it to supp. material. Same for parts of Fig. 2, Fig. 4, as well as Fig. 8

-Figures have been changed accordingly. Now there are only seven figures instead of eleven. Rest of the important figures have been moved to supplementary material.

11. correct question marks in figure designations three times (and also in the discussion)

- Corrected

Query 12 (Discussion)

Consider labeling the sequences of the alignment with the MazE type and instead write the PDB accession numbers into the figure legend

The changes have been made accordingly. The change can be visualised in the Manuscript Fig. 3F.

Miscellaneous

There is a number of typos in the text and also in the References (e.g. when umlauts are used), please check carefully

We thank the reviewer for pointing out the errors. We have corrected the errors in the text as well as in the references.

References

- [1] Prabhakar Tiwari, Garima Arora, Mamta Singh, Saqib Kidwai, Om Prakash Narayan, and Ramandeep Singh. Mazf ribonucleases promote mycobacterium tuberculosis drug tolerance and virulence in guinea pigs. *Nature communications*, 6(1):1–13, 2015.
- [2] Longxuan Zhao and Junjie Zhang. Biochemical characterization of a chromosomal toxin–antitoxin system in mycobacterium tuberculosis. *FEBS letters*, 582(5):710–714, 2008.
- [3] Gopinath Chattopadhyay, Munmun Bhasin, Shahbaz Ahmed, Tannu Priya Gosain, Srivarshini Ganesan, Sayan Das, Chandrani Thakur, Nagasuma Chandra, Ramandeep Singh, and Raghavan Varadarajan. Functional and biochemical characterization of the mazef6 toxin-antitoxin system of mycobacterium tuberculosis. *Journal of Bacteriology*, 204(4):e00058–22, 2022.
- [4] Liisa Holm. Using dali for protein structure comparison. In *Structural Bioinformatics*, pages 29–42. Springer, 2020.
- [5] Teresa Cortes, Olga T Schubert, Graham Rose, Kristine B Arnvig, Iñaki Comas, Ruedi Aebersold, and Douglas B Young. Genome-wide mapping of transcriptional start sites defines an extensive leaderless transcriptome in mycobacterium tuberculosis. *Cell reports*, 5(4):1121–1131, 2013.
- [6] Xi Shan, Kevin H Gardner, DR Muhandiram, NS Rao, Cheryl H Arrowsmith, and Lewis E Kay. Assignment of 15n, 13c α , 13c β , and hn resonances in an 15n, 13c, 2h labeled 64 kda trp repressor-

operator complex using triple-resonance nmr spectroscopy and 2h-decoupling. *Journal of the American Chemical Society*, 118(28):6570–6579, 1996.

REVIEWERS' COMMENTS:

Reviewer #2 (Remarks to the Author):

The revised manuscript has addressed my main concerns. It is nevertheless recommended to correct the number of significant digits and errors in the values that correspond to the ITC measurements (Table 2, Figure 6 legend and also in the text).

Reviewer #3 (Remarks to the Author):

The comments of this reviewer to the previous version have now been adequately addressed.

Manuscript ID - COMMSBIO-22-1497-A
Response to Reviewers

Khushboo Kumari and Siddhartha P Sarma*

Response to Reviewer 2

Query

The revised manuscript has addressed my main concerns. It is nevertheless recommended to correct the number of significant digits and errors in the values that correspond to the ITC measurements (Table 2, Figure 6 legend and also in the text).

The manuscript has been revised accordingly.